# Layer V Neocortical neurons from individuals With drug-resistant epilepsy show multiple synaptic alterations but lack somatic hyperexcitability

Luis A. Márquez[1,¤a], Christopher Martínez-Aguirre[1,¤b], Estefanía Gutierrez-Castañeda[1], Ernesto Griego[1,¤b], Isabel Sollozo-Dupont[4], Félix López-Preza[1], Mario Alonso-Vanegas[3], Luisa Rocha Arrieta[1], Emilio J. Galván[1,2*]

1 Departamento de Farmacobiología, Cinvestav Sur, Ciudad de México, México, 2 Centro de Investigaciones sobre el Envejecimiento, CIE, Ciudad de México, México, 3 Hospital HMG Coyoacán, Ciudad de México, México, 4 Instituto Nacional de Perinatología, Isidro Espinosa de los Reyes, Ciudad de México, México

¤a Current address: Department of Psychological and Brain Sciences, University of Wisconsin-Milwaukee, Milwaukee, WI, USA
¤b Current address: Dominick P. Purpura Department of Neuroscience, Albert Einstein College of Medicine, Bronx, NY, USA
* ejgalvan@cinvestav.mx

## Abstract

Although neuronal hyperexcitability is the primary mechanism underlying seizure activity in epilepsy, little is known about how different neuronal mechanisms at different organizational levels contribute to network hyperexcitability in the human epileptic brain. In this study, we determined a series of cellular and synaptic properties of Layer V pyramidal neurons from neocortical tissue of patients with drug-resistant epilepsy that may contribute to the hyperexcitable state associated with epilepsy. Using the whole cell, patch-clamp technique, and extracellular recordings, we determined the passive and active electrophysiological properties of Layer V pyramidal neurons with regular spiking phenotypes from temporal, parietal, and frontal neocortices surgically resected from individuals with drug-resistant epilepsy. Also, the glutamatergic strength, the synaptic coupling between presynaptic volleys and field excitatory postsynaptic potentials, and short-term, frequency-dependent plasticity were determined. Our data revealed that pyramidal neurons exhibit minimal spontaneous synaptic activity, similar resting membrane potentials, and input resistance values among the temporal, parietal, and frontal neocortices. Although frontal neurons were more hyperexcitable than temporal and parietal neurons, the firing output was comparable to that previously observed in non-pathological human tissue. In contrast, the extracellular recordings uncovered significant decoupling between presynaptic excitability and postsynaptic activity and the lack of short-term depression in response to gamma-range (30 Hz) repetitive stimulation. Our data suggest that neocortical Layer V pyramidal neurons from individuals with drug-resistance epilepsy, particularly

**Data availability statement:** All relevant data are within the manuscript and its Supporting Information files.

**Funding:** This work was supported by financial funding from Conahcyt to LR (A3-S-26782); Cinvestav-IPN to EJG. Additional funding was provided by doctoral fellowships 725800 (LAM) by Conahcyt. The funders had no role in study design, data collection and analysis, decision to publish, or preparation of the manuscript.

**Competing interests:** The authors have declared that no competing interests exist.

intratelencephalic-2 neurons, which exhibit regular firing, are not necessarily hyperexcitable at the somatic level. Instead, synaptic alterations, such as synaptic decoupling and the lack of frequency-dependent short-term depression, may significantly contribute to the hyperexcitable state observed during seizure activity.

## Introduction

Although significant progress has been made in animal models to understand the intrinsic and synaptic alterations that can trigger neuronal hyperexcitability, —a hallmark of epilepsy alongside recurrent seizure activity [1–6],— the translation of these neurophysiological findings to the brain of patients with epilepsy remains largely unverified, with few exceptions [7,8]. In this line of work, neocortical hyperexcitability at the network level is recognized as a central mechanism underlying seizure activity [7,9]. However, the potential contribution of mechanisms operating at lower organizational levels, such as cellular and synaptic mechanisms, remains less well understood [3,10,11].

Conceptually, multiple neuronal mechanisms operating at different organizational levels may contribute to the emergence of network hyperexcitability. At lower levels, such as the cellular and synaptic, alterations in ion channel function and intrinsic membrane properties can modify cellular excitability. In contrast, changes in synaptic interactions within local circuits can alter synaptic drive and plasticity [3,10], thereby influencing the propagation of electrical activity. At higher levels, including the network and system levels, interactions between neural circuits, or even between the nervous and immune systems, may further promote hyperexcitability [3]. Despite significant advances, the precise nature of hyperexcitability remains unknown, raising the question of whether network hyperexcitability in epilepsy results from the convergence of multiple mechanisms across organizational levels or from alterations at only a subset of them.

A more realistic approach to studying the cellular and synaptic mechanisms underlying the hyperexcitable state associated with epilepsy is to directly examine pathological brain tissue from surgical procedures in epileptic patients. Nevertheless, given the limited access and bioethical concerns in obtaining human tissue, these findings are often compared with results from analogous structures in the rodent brain. Furthermore, a growing number of studies have documented numerous differences between rodent and human brain structures. For example, the process of synaptic integration in pyramidal cells (PCs) of the neocortex differs between rodents and humans [12–14]. These subtle differences extend to the functional expression of ion channels, their biophysical properties, neuronal firing, short-term plasticity, and information-processing efficiency [15,16]. Likewise, morphological differences exist between the human and rodent neocortex, including dendritic complexity of pyramidal neurons [17,18]. These differences, along with a limited understanding of human neuronal physiology in its normal state [10,11], may contribute to misinterpretations of the cellular mechanisms underlying hyperexcitability in epilepsy.

Remarkably, Layer V PCs have been identified as a primary source of interictal discharges in epilepsy [19–21], a phenomenon that may be partially explained by their morphological and biophysical properties [22,23], Therefore, this study examined the biophysical and synaptic differences of Layer V PCs in the temporal, parietal, and frontal cortices resected from patients with drug-resistant epilepsy (DRE). Patch-clamp recordings in acute human neocortical slices were used to classify PCs based on their membrane properties and firing patterns, as previously reported [24]. Extracellular recordings assessed synaptic strength, coupling between presynaptic volleys and excitatory postsynaptic potentials, and short-term plasticity in response to gamma-range stimulation. Due to the inherent challenge of obtaining 'healthy' tissue for comparative purposes (see [11]), the findings from this study were interpreted alongside studies using non-pathological tissue. Our results suggest that intrinsic cellular and synaptic mechanisms in Layer V PCs may contribute differentially to the dominant network hyperexcitability observed during seizures in patients with DRE.

## Materials and methods

### Preparation of human cortical slices

Neocortical brain tissue of the temporal, parietal, or frontal neocortex was obtained during surgical procedures of 10 patients with DRE (Table 1). The resected brain samples were obtained directly from epileptic foci identified by intraoperative electrocorticograms, which confirm their hyperexcitable nature at the network level. Tissue samples were collected between June 21, 2022, and July 23, 2023. This study was authorized by the Ethics, Biosecurity, and Scientific Committee of the National Commitment on Scientific Research (CNIC-785) of Instituto Mexicano del Seguro Social under number FR-2019-785-008 that waived the consent from parents of the minors included in the study.

Immediately after resection, the epileptic neocortex was placed in ice-cold isotonic buffer solution (in mM: 320 sucrose, 1 EDTA, 5 Tris-HCl; pH = 7.4; ≈ 315 mOsm) with continuous bubbling of carbogen (95% $O_2$/5% $CO_2$ at 0.5 L/min). The transportation time between the resection and the beginning of slice preparation in the neurophysiology laboratory was less than 45 minutes. The experimental procedures (protocols: 055/2018 and 055/2019) were approved by our institution's internal ethics committee. The procedures for obtaining brain tissue from epileptic patients (protocol FR-2019-785-008) were approved by the internal ethics committees of the involved medical institutions, in accordance with the Declaration of Helsinki. Written informed consent was obtained from all patients. Parental consent was obtained for all minor participants. Patients' clinical variables, such as sex, age, epileptic focus location, and drug treatment, are summarized in Table 1.

**Table 1. Clinical variables of patients with drug-resistant epilepsy.**

| ID | Sex | Age (years) | Age of seizure onset (years) | Duration of epilepsy (years) | Frequency of the seizures (per month) | Location of the focus | ASMs administered before surgery |
|---|---|---|---|---|---|---|---|
| **HUM-209** | F | 21 | 10 | 11 | 7 | Left temporal lobe | LEV, OXC, NMP |
| **HUM-210** | F | 30 | 13 | 17 | 20 | Right frontal lobe | OXC, LEV, TOP |
| **HUM-211** | M | 21 | 13 | 8 | 12 | Right temporal lobe | LEV, VPA, PAL, OXC, ZSM |
| **HUM-214** | M | 2.6 | 1.2 | 1.4 | 10 | Right frontal lobe | LEV, LAC |
| **HUM-221** | F | 27 | NA | NA | NA | Right temporal lobe | LEV, LAC, PAL, OXC |
| **HUM-215** | M | 16 | 10 | 6 | 150 | Left frontal lobe | LAC, LEV, CLB, CZM, VAP, LAM |
| **HUM-216** | F | 2 | 0.9 | 1.1 | 3 | Left parietal lobe | LEV, CLB, VAP, LAM, OXC |
| **HUM-217** | F | 7 | 1.25 | 5.75 | NA | Left parietal lobe | LEV, OXC, BRI, PAL, LAC, VPA, CLB |
| **HUM-218** | F | 24 | 16 | 8 | NA | Right frontal lobe | LAM, VAP, LEV |
| **HUM-219** | M | 4 | 2.75 | 1.25 | 1 | Right parietal lobe | VPA |

ASMs, Antiseizure medications; BRI, brivaracetam; CBZ, carbamazepine; CLB, clobazam; CZM, clonazepam; F, female; LAC, lacosamide; LAM, lamotrigine; LEV, levetiracetam; M, male; NA, not available; NMP, nimodipine; OXC, oxcarbazepine; PAL, perampanel; TOP, topiramate; VPA, valproic acid; ZSM, zonisamide.

In the laboratory, brain tissue samples were transferred to a chilled sucrose solution while carbogen was maintained. The sucrose solution composition was (in mM; $\approx$ 290 mOsm): 210 sucrose, 2.8 KCl, 2 $MgSO_4$, 1.25 $Na_2HPO_4$, 25 $NaHCO_3$, 1 $MgCl_2$, 1 $CaCl_2$, and 10 D-glucose. Then, coronal slices of the human neocortex (350 μM thickness) were obtained from neocortical tissue blocks dissected from the tissue samples. The neocortical tissue blocks were sliced perpendicular to the pial surface using a vibratome (Leica VT1000S; Nussloch, Germany). Next, the acute slices were maintained at 34°C for 25–30 minutes in artificial cerebrospinal fluid solution (ACSF; pH $\approx$ 7.30–7.35; $\approx$ 315–320 mOsm) with the following composition (in mM): 125 NaCl, 2.5 KCl, 1.25 $Na_2HPO_4$, 25 $NaHCO_3$, 4 $MgCl_2$, 1 $CaCl_2$, and 10 D-glucose. Then, the slices were maintained at room temperature for at least 1 hour before whole-cell patch-clamp and extracellular recordings were performed. For the recordings, an extracellular solution (modified ACSF) was continuously perfused to slices containing (in mM; $\approx$ 290–295 mOsm): 125 NaCl, 2.5 KCl, 1.25 $Na_2HPO_4$, 25 $NaHCO_3$, 1.5 $MgCl_2$, 2.5 $CaCl_2$, and 10 D-glucose. All recordings were performed at 32.5 $\pm$ 1°C.

## Whole-cell recordings

The PCs were visualized with infrared differential interference contrast optics coupled to an FN1 Eclipse microscope (Nikon Corporation, Minato, Tokyo, Japan). PCs located in Layer V were identified based on their characteristic shape (broad base, pointed apex, and a single dendritic elongation pointing towards the pia-cortex) and position. The patch pipettes were pulled from borosilicate glass using a micropipette puller (P97, Sutter Instruments, Novato, CA, USA). The pipette tips had a resistance of 4–6 MΩ when filled with an intracellular solution (pH $\approx$ 7.20–7.28, osmolarity $\approx$ 315–325 mOsm/L) with the following composition (in mM): 135 $K^+$-gluconate, 10 KCl, 5 NaCl, 1 ethylene glycol-bis($\beta$-aminoethyl ether)-N,N,N′,N′-tetraacetic acid (EGTA), 10 N-(2-hydroxyethyl)piperazine-N′-(2-ethane sulfonic acid) (HEPES), 2 $Mg^{2+}$-ATP, 0.4 $Na^+$-GTP, and 10 phosphocreatine. Biocytin (0.4%) was routinely added to the intracellular solution for post hoc digital reconstructions and morphological analysis of the recorded neurons. Whole-cell patch-clamp recordings were performed using an Axopatch 200B amplifier (Molecular Devices, San José, CA, USA), digitized at 40 kHz, and filtered at 1 kHz with a Digidata 1550B (Axon Instruments, Palo Alto, CA, USA). Digital signals were acquired and analyzed offline using pCLAMP 11.2 (Molecular Devices).

## Determination of passive and active electrophysiological properties

The resting membrane potential (RMP) was measured after the initial break-in from giga-seal to whole-cell configuration in current-clamp mode, followed by the determination of spontaneous synaptic activity (SSA), which was monitored for 3 minutes in gap-free mode. Next, the cells were held at RMP, and a series of hyperpolarizing and depolarizing current pulses (from −300 pA to 60 pA; 30 pA steps, 1s of duration) were somatically injected to determine the current-voltage (I–V) relationship, somatic input resistance ($R_N$), membrane time constant ($\tau$), and membrane capacitance. $R_N$ determined as the slope of a linear fit ($f(x) = mx + b$) to the steady-state I–V plot elicited by subthreshold current injection (−60–0 pA). $\tau$ was calculated by fitting a single exponential function ($f(t) = \sum_{i=1}^{n} Ai\, e^{-\frac{t}{\tau i}} + C$) to a voltage response elicited by a current pulse of −30 pA. Membrane capacitance (pF) was calculated as $\tau / R_N$. Next, cells were held at −70 mV, and the rheobase and latency to the first action potential (AP) were measured using depolarizing current ramps (30 pA steps, 500 ms duration). The AP kinetic measurements included half-width (H-W), maximal depolarization slope (MDS), and maximal repolarization slope (MRS). These kinetic parameters were depicted in phase plots, constructed by plotting the first derivative of membrane potential, dV/dt (mV/ms), as a function of membrane potential. The firing threshold at rheobase was defined as the membrane potential at which dV/dt exceeded 15 mV/ms. To determine the recorded cells' excitability level, a series of firing rate–current curves were constructed by computing the number of elicited APs in response to increasing current injection from 0 to 390 pA (30 pA steps; 1s duration). From these curves, the neuronal gain was calculated as the slope of a linear fit to the steady-state rate firing–current plot obtained by current injection from 90 to 210 pA, which showed null or sparse firing adaptation. Series resistance was monitored throughout the recording and ranged from 14 to

25 MΩ. Cells that exhibited a > 20% change were excluded from the analysis. The uncorrected liquid junction potential was 12–13 mV.

## Classification of neocortical cells according to firing pattern

PCs' electrophysiological identity was determined by visualizing neuronal firing patterns and analyzing spike frequency adaptation [25]. Spike frequency adaptation was calculated as the slope of the linear fit to the relationship between the interspike interval (ms) and the time of occurrence (ms) of the second spike in each pair of consecutive spikes, elicited by a 210 pA current pulse (1 s). The slope values revealed that PCs exhibited three distinct firing patterns: regular spiking (RS), intrinsic bursting, and adaptive firing. In addition, the instantaneous firing frequency (IFF) was computed as the reciprocal value of the ISI during consecutive APs. These data were plotted as the instantaneous firing frequency of each AP elicited by increasing current pulses (120–360 pA; 1 s) for each neuronal firing type.

## Extracellular recordings

Extracellular recordings were used to analyze the synaptic strength. A bipolar stimulation electrode was placed in Layer I/II, and the resulting field excitatory postsynaptic potential (fEPSP) was recorded in Layer Va with a borosilicate pipette (resistance: 1–2 MΩ when filled with 3M NaCl solution). The current pulses were delivered via a high-voltage isolation unit (A365D; World Precision Instruments, Sarasota, FL, USA) under the command of a Master-8 pulse generator (AMPI, Jerusalem, Israel). Electrical responses were amplified using a Dagan BVC-700A amplifier (Minneapolis, MN, USA) connected with a 100x gain headstage (Dagan, model 8024) and a high-pass filter set at 0.3 Hz. Additional electrical noise suppression was achieved using a Humbug device (Quest Scientific Instruments; North Vancouver, BC, Canada). The evoked fEPSPs were displayed on a computer-based oscilloscope and digitized with an A/D converter (BNC-2110) for storage and offline analysis with LabVIEW 7.1 software (National Instruments, Austin, TX, USA).

## Characterization of fEPSP Waveform, Paired-Pulse Ratio, Frequency-Dependent Short-Term Plasticity, and Neuronal Coupling

The fEPSPs were acquired at 0.067 Hz with current pulses (200–250 µA) of 100 µs duration. Ten consecutive sweeps were averaged, then analyzed to measure the kinetic parameters of the fEPSP waveform. For each fEPSP, the amplitude (measured in mV), H-W (ms), peak time (ms), fiber volley (FV) amplitude (mV), and slope (mV/ms from 10% to 80% of response) were measured with custom-made software written in LabVIEW 7.1 or pCLAMP 11.2 software.

For paired-pulse ratio (PPR) analysis, 12 consecutive synaptic responses with paired stimulation (inter-stimulus interval of 60 ms; 15%–20% of maximal response) were acquired at 0.067 Hz. The PPR was calculated as the ratio of the second response's amplitude to the first response's amplitude. To determine short-term plasticity in response to brief high-frequency stimulation, the fEPSP baseline response was set to 50% of its maximal amplitude and then recorded for 10 min, as previously reported [26]. Then, a train of 10 current pulses (100 µs in duration) at 30 Hz was delivered in Layers I and II. The train's evoked synaptic responses were normalized to the slope of its first synaptic response and plotted as the number of stimuli vs. normalized fEPSP slope (%). Additionally, spontaneous postsynaptic potentials were detected and quantified during train stimulation. These events occurred more than 10 ms after electrical stimuli and were therefore considered spontaneous, likely generated by short-term synaptic properties rather than evoked activity. The spontaneous postsynaptic potentials were detected using pCLAMP 11.2 software according to the following criteria: an inward-directed waveform, an amplitude > 0.15 mV, and a duration > 1 ms.

To establish the relationship between presynaptic action potentials (fiber volleys, FV) and fEPSPs in the neocortical region, a Pearson correlation analysis ($r^2$) was used. The least squares method was applied to find the equation of the best-fitting curve or line to the dataset, considering the following formula: $y = mx + b$, where $y$ is the dependent variable (fEPSP), $x$ is the independent variable (FV), $m$ is the slope of the line, and $b$ is the $y$-intercept. As a reference group for

normal synaptic coupling, we performed a similar analysis using previously published data from Layer V of the rat pre-frontal cortex [27]. The formulas to calculate the slope ($m$) and intercept ($b$) of the line are derived from the following equations:

$$m = \frac{n(\sum xy) - (\sum x)(\sum y)}{n(\sum x^2) - (\sum x)^2}; \quad b = \frac{(\sum y) - m(\sum x)}{n},$$ where n is the number of data points, $\sum xy$ is the sum of the product of each pair of $x$ and $y$ values, $\sum x$ is the sum of all $x$ values, $\sum y$ is the sum of all y values, and $\sum x^2$ is the sum of the squares of $x$ values.

The observed values of $x$ were used in conjunction with the formula $y = mx + b$ to calculate the predicted values of $y$. The residual values were estimated as the difference between the observed $y$ values and the predicted $y$ values. Lastly, an exploratory analysis (EA) of the residuals was conducted to assess the feasibility of using the least squares method. The EA used the Kolmogorov–Smirnov test, which assesses whether the residuals follow a specific distribution, thereby helping us evaluate the normality assumption required by the least squares method. Additionally, the biases of the residuals were determined using a one-sample Student's t-test. This test allowed us to evaluate whether the residuals' mean deviated significantly from zero, helping us identify any systematic bias in the model's predictions. A non-significant result would suggest that the model's predictions were unbiased.

## Immunostaining of biocytin-filled neurons, confocal image acquisition, and analysis of dendritic complexity

Slices containing the biocytin-filled neurons obtained from the patch-clamp recordings were fixed in 4% PFA solution in 0.1 M PB (pH = 7.4) and kept at 4°C for 24–48 hours for fixation. Then, the slices were washed with PBS and incubated in a blocking/permeabilization buffer (PBS containing 3% BSA and 0.3% Triton X-100) at room temperature for 2 hours. Subsequently, the slices were incubated overnight at room temperature with Streptavidin Texas Red (diluted 1:500, Vector Laboratories; SA-5006). Finally, the sections were washed again and mounted using a Vectashield Vibrance curing antifade mounting medium (Vector Laboratories; H-1800). Photomicrographs were acquired using confocal microscopy (LSM 800 with Airyscan; Carl Zeiss) [28]. The excitation wavelength for Texas Red was 561 nm. Multiple tile z-stack series were captured from each cell and reconstructed using Zen 2.3 (Carl Zeiss) and ImageJ (NIH Image) software.

Dendritic arborization was quantified in four neurons that exhibited staining in the somatic region and dendritic arborizations. Dendrites were quantified by counting the number of total and primary dendrites, the total dendritic length, the axon extension, and the total projection area using ImageJ software coupled with the Sholl analysis plug-in v3.4.2 for morphometric analysis [29,30]. Primary neurites were defined as those that originated directly from the soma. Sholl analysis was used to determine the complexity of dendritic branching. Concentric circles were drawn every 10 µm from the soma, and the number of intersections between the increasing circles and the dendritic arbor was quantified, as well as the number of branch points and end tips. The axon's initial segment was identified morphologically as the point at which the axon's diameter became constant, lacking further tapering or structural specialization, and where the axon exhibited a uniform caliber and trajectory without visible varicosities or branching, as previously reported [17,31].

## Data analysis

During the experimental procedures, the experimenters were blinded to both the tissue brain region of origin and the patients' clinical variables. All data are numerically expressed as mean ± SD unless otherwise stated. The normality of the data was assessed using the Shapiro-Wilk test, which is suitable for small sample sizes (n > 3). No outliers were removed from the data. The group size per cortex type is expressed as the number of cells or slices obtained from $n$ patients. The comparability among experimental conditions was assessed by a ratio-paired Student's t-test, one-way analysis of variance (ANOVA), or mixed-effects ANOVA, as appropriate. In the mixed-effects ANOVA, we analyzed cortex type and current injection as between-subjects and within-subjects factors, respectively. Tukey's post hoc test was computed for multiple comparisons among experimental groups only when F achieved minimal statistical significance. For all the experiments, data were considered significant if $p < 0.05$.

## Results

### Identification of neocortical pyramidal cells by firing pattern from patients with drug-resistant epilepsy

We recorded 34 PCs in acute slices of the human neocortex (Fig 1A) using the whole-cell patch-clamp configuration; the intracellular solution routinely included biocytin for intracellular labeling. The tissue was surgically resected from 10 patients with DRE ranging from 2 to 30 years old of both sexes (Fig 1B, upper panel and Table 1). The average time between the surgical resection, brain slice preparation, and slice placement in the recording chambers was $3 \pm 0.5$ h. According to its origin, the tissue was obtained from the temporal, parietal, and frontal neocortices (Figs 1A and 1B, middle panel).

The somata of PCs were positioned in the middle region of Layer V of the neocortex (Fig 1C), 50–150 μm below the slice surface. Under DIC microscopy, the PC somata size was $22.2 \pm 4.3$ μm, and was visibly different from local interneurons that exhibited smaller, rounded, or bifurcated somata. Strictly pyramidal cell-like morphology, including a broad base, a pointed apex, and the protrusion of at least one major dendritic branch pointing towards the cortex-pia, was an inclusion criterion for patch-clamp recordings. Following the initial membrane break-in from giga-seal to whole-cell configuration, we switched to the current-clamp mode, and SSA was monitored at the RMP of the recorded cells. Since the tissue was obtained from epileptic individuals, the SSA was a parameter routinely used to identify possible hyperexcitable cells. However, PCs in this study showed sparse SSA at their RMP (right insets, Fig 1D).

In the human neocortex, Layer V PCs are classified as either extratelencephalic (ET) neurons or intratelencephalic (IT, types 1 and 2) neurons based on their transcriptomic and electrophysiological profiles [24]. Therefore, we first classified the PCs according to their firing pattern (Fig 1D). Independent of the origin of the epileptic tissue, we found that 22 (64.71%, n = 22 cells/ 10 samples of 10 patients) PCs exhibited a RS firing pattern phenotype, which corresponds to IT-2 neurons. The IT-2 neurons are characterized by stable AP firing, weak AP adaptation, and stable AP amplitude. The other 10 (29.41%, n = 10 cells/ 4 samples of 10 patients) PCs exhibited an intrinsic bursting firing pattern, which corresponds to ET neurons, and are characterized by clustered spikes (2–3) observed during depolarizing plateau potentials and followed by quiescent periods. The remaining 2 (5.88%, n = 2 cells/ 2 samples of 10 patients) PCs exhibited adaptive firing, consistent with the IT-1 neurons, characterized by a single AP followed by a defined afterhyperpolarization, and strong spike adaptation was recorded (Fig 1D, lower panel). Since the cells in this study did not exhibit SSA, we assumed that the firing patterns were intrinsically regulated and not derived from synaptic activity impinging on the recorded PCs, as summarized in the voltage traces and the accompanying SSA in Fig 1D.

To corroborate the neuronal identity of PCs as ET (n = 10 cells/ 4 patients) or IT-2 (n = 22 cells/ 10 patients) neurons, we compared electrophysiological properties that distinguish these neuronal populations [24], which were more abundant in our study. The corresponding measurements for IT-1 neurons are provided in the supplementary information. First, we compared firing rates in response to increasing current injections (0–390 pA, 30 pA steps). However, no significant differences were observed in the firing rate curves between IT-2 and ET neurons (maximal firing rate in IT-2 neurons = $23 \pm 7.86$ Hz, in ET neurons = $21 \pm 7.72$ Hz, unpaired t-test $t_{(30)}$ = 0.97, $p = 0.34$; Fig 1E), and similarly, no differences were observed in neuronal gain (gain in IT-2 neurons = $0.089 \pm 0.04$ Hz.pA$^{-1}$, in ET neurons = $0.07 \pm 0.05$ Hz.pA$^{-1}$, unpaired t-test $t_{(30)}$ = 1.128, $p = 0.268$; inset in Fig 1E). However, as expected, ET neurons exhibited a considerably higher instantaneous frequency firing compared to IT-2 (IFF in ET neurons = $99.62 \pm 26.94$ Hz; in IT-2 neurons = 44.1 26.96 Hz, unpaired t-test $t_{(30)}$ = 5.849, $p > 0.001$).

Next, to corroborate differences in the output patterns of these neuronal populations, we analyzed spike adaptation during current injection (210 pA, 1 s; Fig 1G), a phenomenon in which the ISI gradually increases over time in response to a continuous current stimulus [25]. According to this analysis, the IT-2 neurons showed significantly lower slope values than ET neurons, corroborating their spare adaptive nature (slope in IT-2 neurons = $0.031 \pm 0.028$, in ET neurons = $0.104 \pm 0.09$, unpaired t-test t(30) = 3.552, $p = 0.0013$; inset Fig 1G).

Given that ET and IT-2 neurons also exhibit differences in the input resistance and functional expression of HCN channels [16,24], we examined whether these differences were present in these neuronal populations. As expected, IT-2

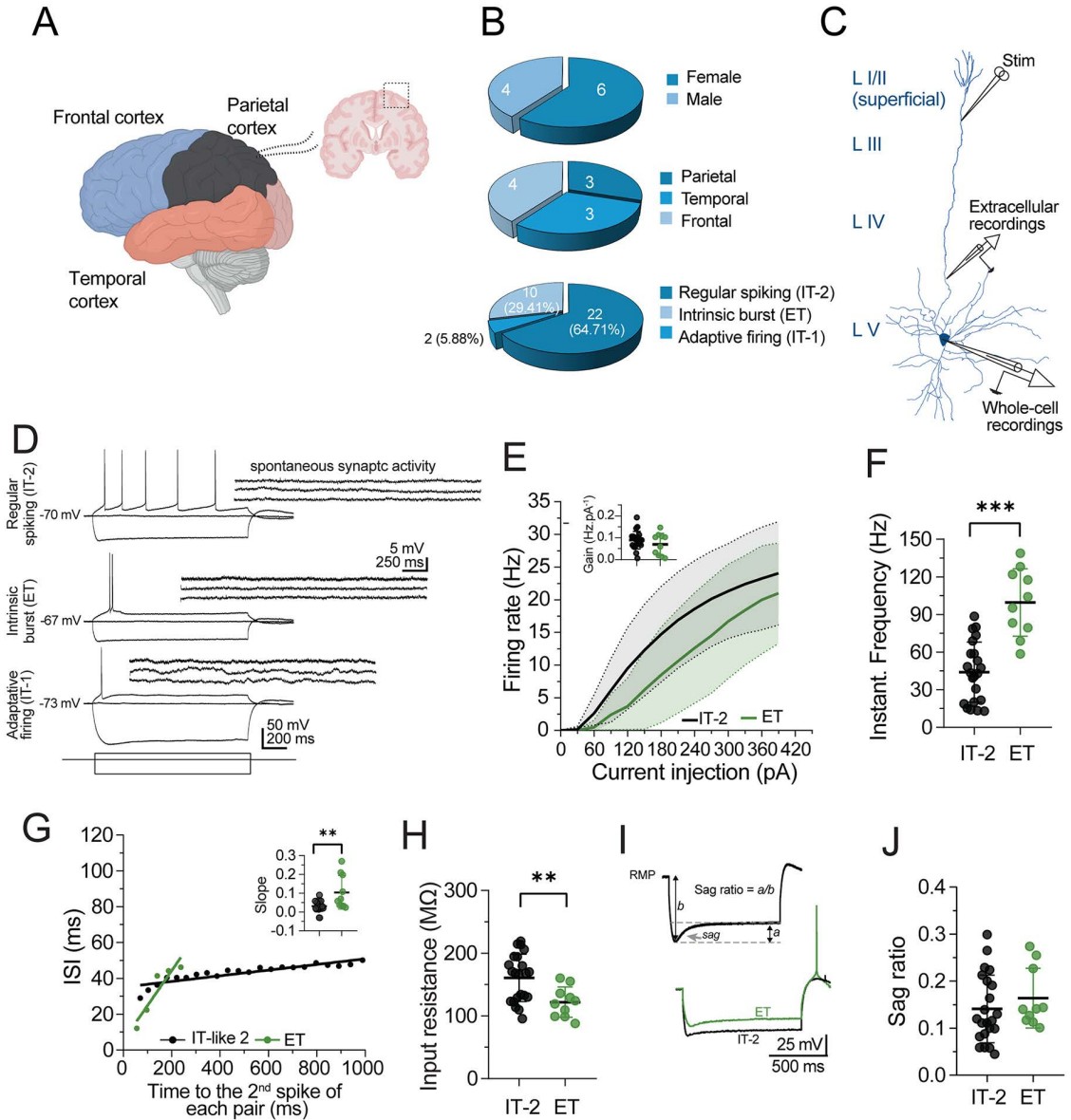

**Fig 1. Layer V pyramidal cell types from the human neocortex with drug-resistant epilepsy.** (A) Schematic representation showing the cortical sections used for this study. Layer (L). Acute cortical slices were prepared from temporal, parietal, and frontal cortices resected from drug-resistant epileptic brains. (B) Pie charts with the percentual distribution of the examined brain tissue (n = 10 patients) by sex (top panel), cortex type (middle panel), and neuronal firing type (lower panel, n = 34 cells). (C) Schematic representation for the whole-cell patch clamp and extracellular recordings in PCs of human cortical slices. The patch pipette was placed in Layer V PCs for whole-cell recordings. For extracellular recordings, the stimulation electrode was placed in the superficial Layers I/II, and synaptic responses were recorded in the deep Layer V. (D) Representative examples of whole-cell recordings showing the firing patterns recorded from PCs. The upper left panel shows a RS neuron or IT-2; the middle panel shows an intrinsic bursting neuron or ET neuron, and the lower panel shows the adaptive firing neuron or IT-1 neuron. A common characteristic of the neurons in this study was their low spontaneous synaptic activity at each neuron's resting membrane potential (as indicated by the calibration bar). (E) Firing rate of IT-2 and ET neurons in response to increasing current injections. The inset compares the gain between groups. (F) Comparison of instantaneous frequency in IT-2 and ET neurons, *** p < 0.001, unpaired t-test. (G) A representative scatter graph of the latency of the second spike of each pair (ms) vs. inter-spike interval (ISI, ms). Note the spare adaptation of IT-2 neurons compared to ET neurons. The inset shows higher slope values in ET neurons than in IT-2 neurons, **p < 0.01, unpaired t-test. (H) Dot plot with error bars showing the differences in input resistance between IT-2 and ET neurons, p ** < 0.01, unpaired t-test. (I) Representative voltage traces evoked by −300 pA (1 s) exemplifying the estimation of sag ratio in IT-2 (black trace) and ET (green trace) neurons. (J) IT-2 and ET neurons did not show differences in sag ratio. n = 22 cells/ 10 samples of 10 patients for IT-2 neurons and n = 10 cells/ 4 samples of 10 patients.

neurons showed higher input resistance values than ET neurons (input resistance in IT-2 neurons = 160.9 ± 37.77 MΩ, in ET neurons = 122 ± 24.1 MΩ, unpaired t-test $t_{(30)}$ = 2.976, $p$ = 0.005; Fig 1H). However, when comparing the sag ratio (see Fig 1H), which is a typical indirect measurement of HCN channel function, no significant differences were found between these populations (sag ratio in IT-2 neurons = 0.141 ± 0.072, in ET neurons = 0.164 ± 0.063, unpaired t-test $t_{(30)}$ = 0.863, $p$ = 0.394; Fig 1J). Together, these results indicate that IT-2 neurons and ET neurons from patients with drug-resistant epilepsy can be distinguished by their firing patterns and electrophysiological properties, such as instantaneous firing frequency, spike adaptation, and input resistance.

## Layer V pyramidal cells in the frontal neocortex exhibit higher firing rates relative to temporal and parietal neurons of patients with drug-resistant epilepsy

Our previous results show that ≈65% of recorded neurons belong to the IT-2 class, characterized by the RS phenotype. Therefore, we contrasted I-T2 neurons' passive and active electrophysiological properties by neocortex types, such as RMP, $R_N$, and τ. Fig 2A shows a representative neuron filled with biocytin, and Fig 2B1–B3 shows representative I–V curves from the different neocortices with the characteristic RS phenotype. Fig 2C summarizes the values of RMP between the three different neocortices (RMP in temporal neurons = −72.3 ± 3.3 mV, n = 6 cells/3 patients; in parietal neurons = −73.4 ± 1.94 mV, n = 5 cells/3 patients; in frontal neurons = −73.13 ± 4.1 mV, n = 11 cells/4 patients). No statistical difference between groups was found in RMP. Next, we compared $R_N$ and τ. However, neither $R_N$ nor τ exhibited significant differences between neuronal groups ($R_N$ in temporal neurons = 164.9 ± 43.65 MΩ; in parietal neurons = 163.9 ± 56.52 MΩ; in frontal neurons = 157.3 ± 27.15 MΩ; Fig 2D; τ in temporal neurons = 26.64 ± 8.93 ms; in parietal neurons = 32.7 ± 4.66 ms; in frontal neurons = 31.91 ± 5.92 ms; Fig 2E). We also determined the membrane capacitance of cells from the different cortices. However, we did not find differences between groups (membrane capacitance in temporal neurons = 166.4 ± 57.37 pF; in parietal neurons = 200.7 ± 24.71 pF; in frontal neurons = 214.5 ± 85.23 pF), suggesting that Layer V PCs across cortical regions in epileptic tissue have similar neuronal surfaces.

Next, we applied a depolarizing current ramp and determined the rheobase and the latency to elicit an AP in IT-2 neurons from the distinct neocortex. Temporal and parietal IT-2 neurons did not show differences (Fig 1F, left and middle panels); however, frontal IT-2 neurons required lower current values to elicit an AP (rheobase in temporal neurons = 166 ± 85.45 pA; in parietal neurons = 101 ± 72.31 pA; in frontal neurons = 65.9 ± 33.1 pA; one-way ANOVA: $F_{(2, 19)}$ = 5.405, Tukey's test, $p < 0.05$; Fig 2F, right panel). On the other hand, we did not detect differences in latency to the AP at the rheobase current between groups (latency in temporal neurons = 416 ± 42.23 ms; in parietal neurons = 385 ± 84 ms; in frontal neurons = 416.6 ± 75.37 ms; Fig 2G).

Next, we compared the firing rate of IT-2 neurons in response to increasing current injection (0–390 pA). The analysis revealed that the maximal firing rates were similar for parietal and temporal neurons (18.2 ± 4.57 Hz vs. 18.4 ± 3.97 Hz, respectively; see black and blue traces, Fig 2H). In sharp contrast, IT-2 neurons from the frontal neocortex exhibited a higher firing rate, reaching a maximal value of 28.82 ± 7.87 Hz (mixed-effects ANOVA, neocortex type effect: $F_{(2, 19)}$ = 5.598, followed by Tukey's test, $p < 0.001$ in frontal neurons vs. temporal and parietal neurons; Fig 2H–I). Consistent with these findings, when computing the neuronal gain, we corroborated the higher firing frequency of IT-2 neurons from the frontal neocortex compared with the temporal and parietal neocortex (neuronal gain in temporal neurons = 0.062 ± 0.037 Hz.pA$^{-1}$, in parietal neurons = 0.067 ± 0.01 Hz.pA$^{-1}$, in frontal neurons = 0.114 ± 0.035 Hz.pA$^{-1}$; one-way ANOVA, $F_{(2, 19)}$ = 6.433, followed by Tukey's test, $p < 0.05$; inset in Fig 2I).

Lastly, we examined the kinetic parameters of the AP waveform elicited at rheobase (Figs 2G and J1), including MDS, MRS, H-W, and firing threshold. Frontal IT-2 neurons exhibited higher MDS values compared with temporal and parietal neurons (MDS in temporal neurons = 179 ± 34.92 mV/ms; in parietal neurons = 184.3 ± 28.1 mV/ms; in frontal neurons = 247.2 ± 23.77 mV/ms; one-way ANOVA: $F_{(2, 19)}$ = 15.07, followed by Tukey's test, $p < 0.05$ in frontal vs. temporal and parietal PCs), suggesting higher global Na$^+$ conductances in frontal PCs. By contrast, the comparison of the MRS,

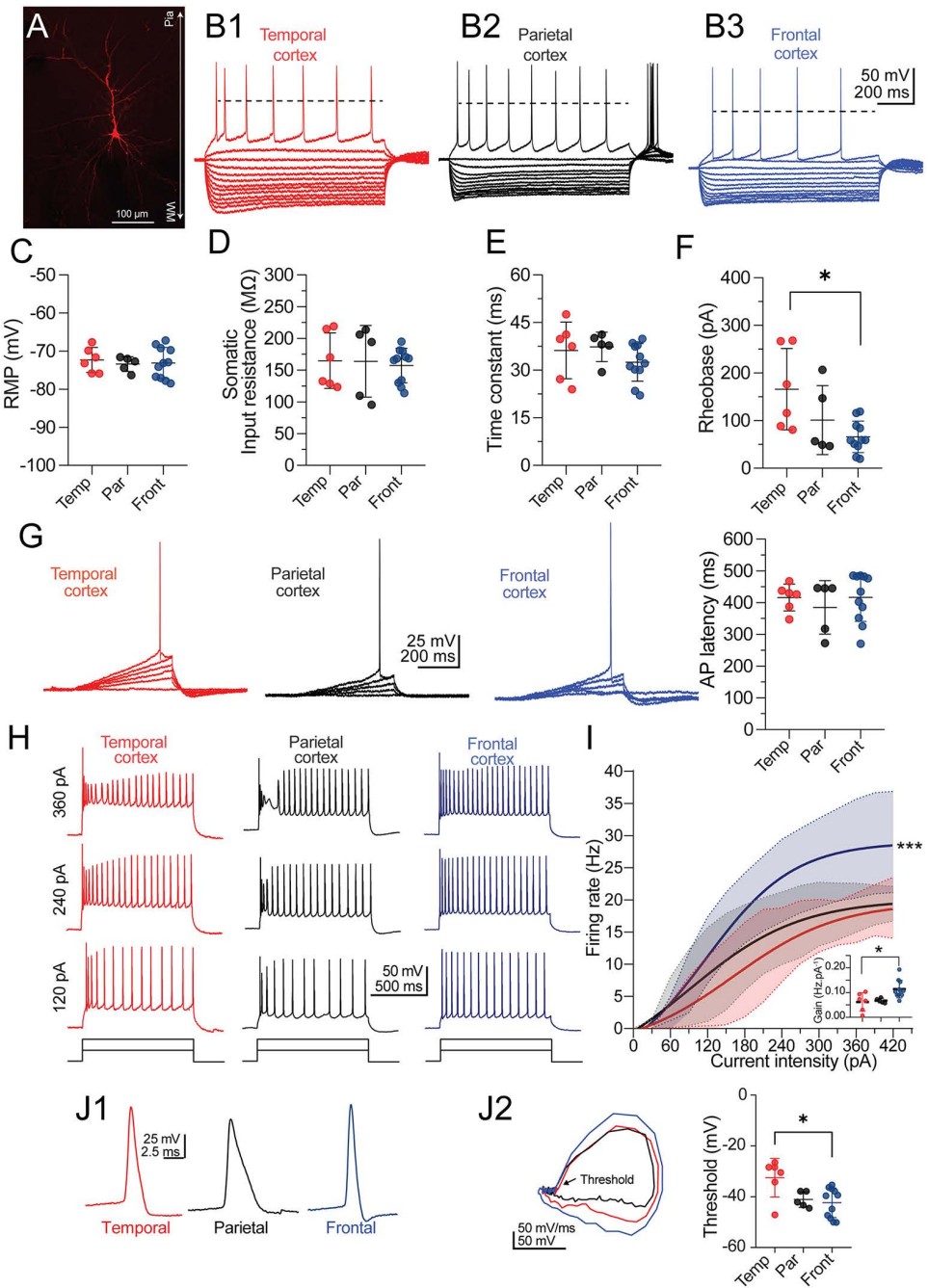

**Fig 2. Passive membrane properties and neuronal firing in IT-2 neurons from temporal, parietal, and frontal cortices of patients with drug-resistant epilepsy. (A)** A typical PC filled with biocytin from the Layer V temporal cortex. WM = white matter. **(B1–B3)** Representative voltage traces of RS PCs of the temporal cortex (B1, n = 6 cells/3 patients), parietal cortex (B2, n = 5 cells/3 patients), and frontal cortex (B3, n = 11 cells/4 patients). The I–V curves were elicited with current injections from −300 pA to 60 pA (30 pA steps). The horizontal dashed line represents the AP overshot. Passive and active parameters obtained from RS neurons: the dot plots with error bars contrast **(C)** the resting membrane potential (RMP), **(D)** the somatic input resistance ($R_N$), and **(E)** the membrane time constant (τ) between PCs of the different cortices. **(F)** Dot plot with error bars summarizing the rheobase current required to elicit one AP by injecting a depolarizing current ramp, as illustrated in Fig 2G. Values for each neuronal group. * $p < 0.05$, one-way ANOVA followed by Tukey's test. **(G)** In the left panel, neuronal firing in PCs in response to a depolarizing current ramp injection. The membrane potential was set at −70 mV. In the right panel, the violin plot shows the AP latency (in ms) elicited with rheobase current. **(H)** Representative firing traces from temporal (red), parietal (black), and frontal (blue) cortices to depolarizing current steps (1 s). **(I)** Line graph with frequency rate (Hz) – current of the

different cortical neurons. Notice that PCs from the frontal cortex exhibited a higher frequency rate than temporal and parietal neurons. *** $p < 0.001$, mixed-effects ANOVA, cortex type effect followed by Tukey's test. Similarly, the comparison of neuronal gain (Hz pA$^{-1}$), calculated as the slope of the linear portion (90–210 pA) of the frequency rate-current curve, corroborated higher excitability in frontal neurons (Fig 2I, dot plot with error bars inset). * $p < 0.05$, one-way ANOVA followed by Tukey's test. **(J1)** Expanded AP traces elicited with rheobase current by cortex type and its representation as a phase plot (left panel in **J2**). The phase plots were constructed by plotting the membrane voltage (mV) vs. its first derivative (mV/ms). Firing threshold at rheobase by cortical region (right panel in **J2**), showing that frontal neurons exhibit a lower threshold than temporal neurons. * $p < 0.05$, one-way ANOVA followed by Tukey's test.

which reflects the activity of outward K$^+$ conductances mediating the fast repolarization of the AP, showed no statistical differences between the neurons from the different neocortex (MRS in temporal neurons = −83.9 ± 37.1 mV/ms; in parietal neurons = −48.2 ± 1.96 mV/ms; in frontal neurons = −77.5 ± 27.47 mV/ms). Likewise, when comparing the H-W values between groups revealed that parietal IT-2 neurons exhibit increased H-W values compared to temporal and frontal PCs (H-W in temporal neurons = 1.7 ± 0.69 ms; in parietal neurons = 2.7 ± 0.75 ms; in frontal neurons = 1.8 ± 0.49 ms; one-way ANOVA: $F_{(2, 19)}$ = 4.796, followed by Tukey's test, $p < 0.05$ in parietal vs. temporal and frontal PCs). Consistent with their lower rheobase and the absence of differences in intrinsic membrane properties, frontal IT-2 neurons exhibited a lower firing threshold compared to temporal neurons (threshold in frontal: −42.38 ± 5.81 mV; temporal: −32.52 ± 7.61 mV; parietal: −41.12 ± 3.11 mV; one-way ANOVA, $F_{(2, 19)}$ = 5.659, followed by Tukey's test, $p < 0.05$; right panel in Fig 2J2). Additionally, a subset of recorded IT-2 neurons (n = 4 of 22) was reconstructed and analyzed, revealing morphological characteristics of Layer V pyramidal neurons (see Fig 5).

## Kinetic properties of neocortical fEPSPs and synaptic coupling in drug-resistant Epilepsy

In parallel with the patch-clamp experiments, we also performed extracellular recordings in independent slices obtained from the same patients' brain samples to explore the synaptic properties of human neocortex. A stimulation electrode was placed in Layers I/II, and the evoked response was recorded in Layer Va (Fig 1C). Parameters of the fEPSP, including amplitude, peak latency, and H-W, were analyzed (Fig 3A). The panels in Fig 3B1–B3 show representative fEPSPs of the temporal, parietal, and frontal neocortex. In the temporal neocortex (n = 8 slices/3 patients), the fEPSP was significantly larger compared with the parietal (n = 8 slices/3 patients) and frontal (n = 8 slices/4 patients) neocortical responses (fEPSP amplitude in temporal neocortex = 0.96 ± 0.1 mV; in parietal neocortex = 0.6 ± 0.2 mV; in frontal neocortex = 0.63 ± 0.2 mV; one-way ANOVA: $F_{(2, 21)}$ = 9.767, Tukey's test, $p < 0.01$ in temporal vs. parietal and frontal neocortex; Fig 3C). However, neither the peak latency nor the H-W of the fEPSPs was different between groups (peak latency in temporal neocortex = 1.67 ± 0.4 ms; in parietal neocortex = 1.75 ± 0.64 ms; in frontal neocortex = 2.06 ± 0.64 ms; Fig 3D; fEPSP H-W in temporal neocortex = 2.1 ± 0.7 ms; in parietal neocortex = 1.82 ± 0.47 ms; in frontal neocortex = 1.94 ± 0.64 ms; Fig 3E).

During the acquisition of fEPSPs, both the FV (or presynaptic action potentials) amplitudes and the corresponding fEPSPs exhibited trial-to-trial variability (i.e., small FVs and larger fEPSPs or vice versa), despite constant current intensity and stimulation frequency (0.067 Hz). Under physiological conditions, the FV–fEPSP relationship is typically linear and represents a classical measure of synaptic coupling between input and output in neuronal microcircuits, whereas pathological conditions alter this relationship [32,33].

To quantify the relationship between presynaptic excitability and postsynaptic response in cortical slices, we first calculated Pearson's $r^2$ values between individual FV amplitudes and their corresponding fEPSP slopes for each cortical region (Fig 3F1–F3). The resulting coefficients of determination were relatively low, suggesting a disrupted input-output coupling: temporal cortex $r^2 = 0.438$; parietal cortex $r^2 = 0.073$; frontal cortex $r^2 = 0.289$. For comparative purposes, data from control rodent slices (n = 5 slices/3 animals; see additional information) showed a strong correlation ($r^2 = 0.925$; Fig 3F4), consistent with a normal synaptic coupling.

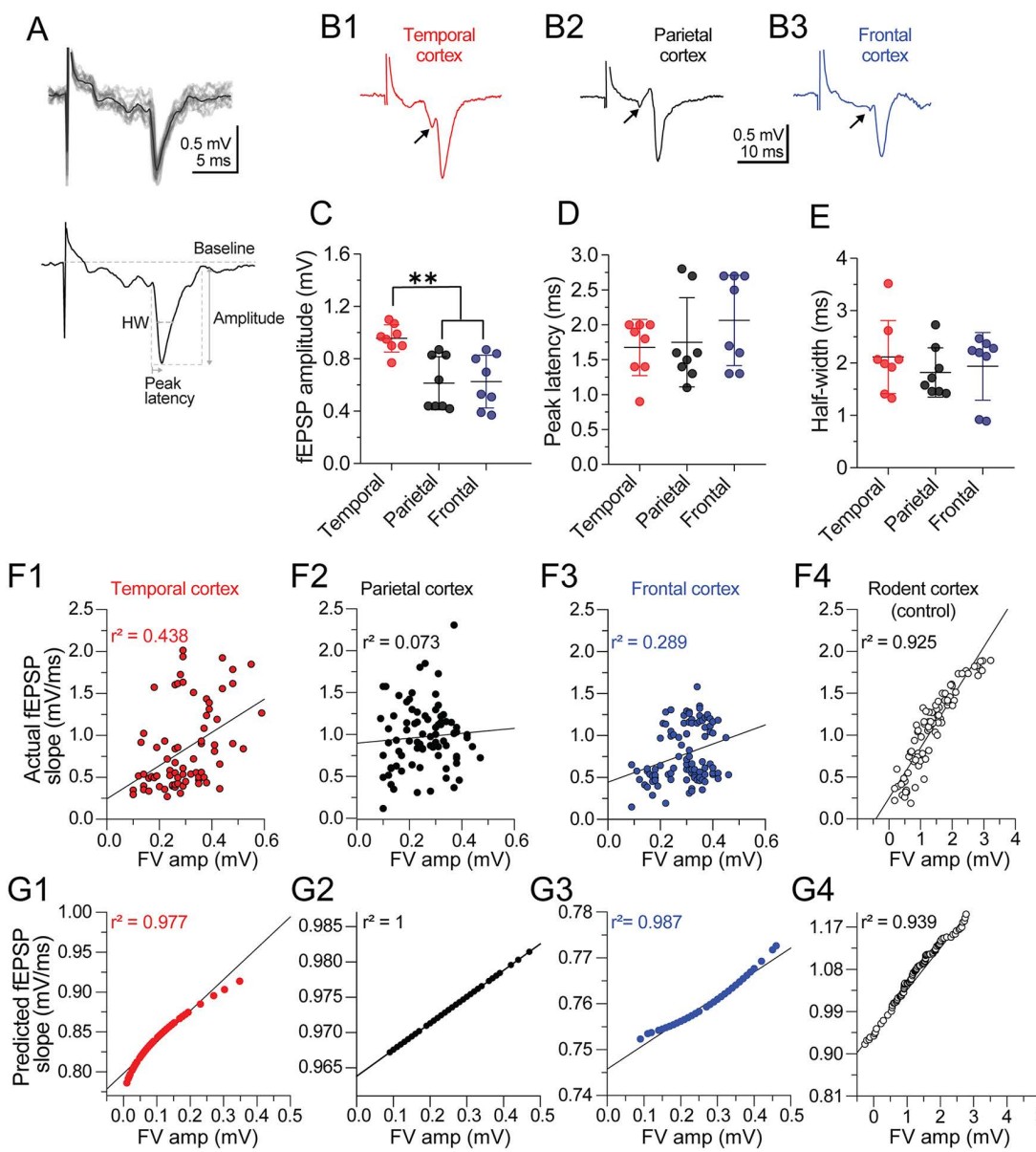

**Fig 3. Kinetic profile of Layer V cortical fEPSPs in patients with drug-resistant epilepsy. (A)** The upper panel is a sequence of 10 consecutive fEPSPs recorded in Layer Va of the cortex in response to electrical stimulation in Layer I/II (see schematic representation in Fig 1C). The individual traces are shown in gray, and the averaged response is in black. The lower panel shows the kinetic parameters derived from the fEPSP waveforms, including fEPSP amplitude (mV), peak latency (ms), and fEPSP half-width (HW, ms). **(B)** Representative fEPSP traces from **(B1)** temporal, **(B2)** parietal, and **(B3)** frontal neocortex. Each trace was averaged from five continuous sweeps acquired at 0.067 Hz. The arrowhead preceding the fEPSP is the presynaptic fiber volley. In the lower panel, a dot plot with error bars summarizes the kinetic parameters of **(C)** the fEPSP amplitude, **(D)** peak latency, and **(E)** fEPSP H-W, obtained from the three cortices. Notice that fEPSP amplitudes from parietal and frontal slices are smaller compared to temporal slices. n = 8 slices/3 patients for temporal and parietal slices; n = 8 slices/4 patients for frontal slices. **p < 0.01, one-way ANOVA followed by Tukey's test. **(F1–F3)** Scatter plot showing the individual relationship between presynaptic fiber volley vs. actual fEPSP amplitude from the three cortical areas and the resulting Pearson's correlation coefficient. Each dot within the plot represents an actual sweep that included FV and fEPSP. **(F4)** Scatter plot showing the same relationship in rodents. Notice the differences in Pearson's *r²* coefficient between DRE and control responses. **(G1–G3)** Scatter plots showing the theoretical consistency and proportionality of the predicted FV vs. fEPSP response for each cortical area according to the least square methods. Under theoretical conditions, cortical responses should be linearly correlated, as occurs in the **(G4)** scatter plot of control responses acquired from rodent brain.

To further characterize the strength of this synaptic coupling, we next performed linear regression analyses of FV and fEPSP slopes using the least-squares method and compared the resulting regression slopes across cortical regions and the control group. The resulting slope magnitude is generally proportional to synaptic coupling efficiency [32–34]. The regression slopes (mean ± SD) were as follows: temporal cortex = 2.88 ± 21.35; parietal cortex = 0.815 ± 27.82; frontal cortex = 1.83 ± 15.33; and control (rodent) group = 5.29 ± 5.68. A t-test revealed that the cortical slopes differed significantly from the control group, supporting the idea of reduced synaptic coupling across cortical regions in patients with DRE.

On the other hand, the linear fits obtained by least-squares analysis showed a strong correlation between FV and fEPSP responses within each cortical region, with slopes indicating partial recovery relative to the control group. As illustrated in Fig 3G1–G3, the coefficients of determination ($r^2$) were remarkably high across all areas: temporal cortex ($r^2 = 0.9541$), parietal cortex ($r^2 = 0.9745$), and frontal cortex ($r^2 = 1.0$). Similarly, the regression slopes were relatively consistent: temporal cortex (m = 4.1), parietal cortex (m = 3.8), frontal cortex (m = 3.8), and control group (m = 5.0). This combination of strong linear correlation and slope recovery suggests that, although the FV–fEPSP relationship may become decoupled under pathological conditions, its fundamental structure remains resilient, a pattern consistent with that observed under basal conditions.

## Neocortical synapses from patients with drug-resistant epilepsy do not exhibit short-term depression

A distinctive property of the neocortical excitatory synapses of rodents is their low probability of neurotransmitter release [35], a phenomenon that governs glutamatergic transmission's PPR. However, PPR has been barely explored in human neocortical synapses. Therefore, the next series of experiments aimed to determine the PPR of the glutamatergic transmission at the Layer I/II – V synapses using an inter-stimulus interval of 60 ms. Figs 4A and 4B show the mild or spare PPR facilitation (PPF) found in the three neocortical areas. In the temporal neocortex, the PPR was 1.03 ± 0.15 (n = 6 slices/3 patients), while in the parietal and frontal neocortex, the PPR was 1.07 ± 0.47 (n = 7 slices/3 patients) and 1.15 ± 0.64 (n = 7 slices/4 patients), respectively (Fig 4B).

According to previous studies, repetitive synaptic stimulation within the gamma-range triggers short-term depression (STD) in rodent and human neocortices [15,35,36]. Therefore, we examined the epileptic neocortex's response to trains of ten stimuli (S) at 30 Hz. To avoid synaptic saturation during the experimental trials, we set the applied current intensity to evoke ≈50% of the maximal fEPSP amplitude in each slice [26]. These responses tended to exhibit a dominant synaptic facilitation that was independent of the neocortex type ($fEPSP_{s10}/fEPSP_{s1} = 127.6 ± 45.54\%$ of S1, paired t-test $t_{(9)} = 1.198$, $p = 0.087$; inset Fig 4D). When distinguished by neocortical area (Fig 4D), the temporal neocortex exhibited sustained facilitation in response to a 30 Hz train (red line), whereas the parietal and frontal neocortex showed unfacilitated or mildly facilitated responses, respectively (black and blue lines). However, the temporal neocortex was the only region with a significant synaptic facilitation in response to 30 Hz train ($fEPSP_{s10}/fEPSP_{s1}$ in temporal neocortex = 162.7 ± 28.87% of S1; ratio-paired t-test, $t_{(2)} = 4.48$, $p = 0.045$; in parietal neocortex = 99.14 ± 29.41% of S1, ratio-paired t-test, $p = 0.814$; in frontal neocortex = 130.6 ± 61% of S1, paired t-test, $p = 0.669$; Fig 4E).

Additionally, we observed that during train stimulation, spontaneous postsynaptic potentials emerged following evoked synaptic responses, with latencies exceeding 10 ms (see arrows in Fig 4C). While the number of spontaneous postsynaptic potentials was significantly higher in temporal neocortex slices compared to parietal slices, no statistically significant difference was found when compared to frontal neocortex slices (events in temporal cortex: 9 ± 2.65; in frontal cortex: 6.67 ± 3.21; in parietal cortex: 2.75 ± 0.96; one-way ANOVA, $F_{(2,7)} = 0.446$, Tukey´s post-hoc test, $p < 0.05$ in temporal vs. parietal cortex; Fig 4F). Interestingly, we found a strong positive correlation between the number of spontaneous postsynaptic potentials and the magnitude of the last synaptic response during the 30 Hz train (Pearson correlation, $r^2 = 0.871$; Fig 4G). These data suggest that frequency-dependent synaptic facilitation or the lack of STD in epileptic neocortex favors excitatory synaptic activity.

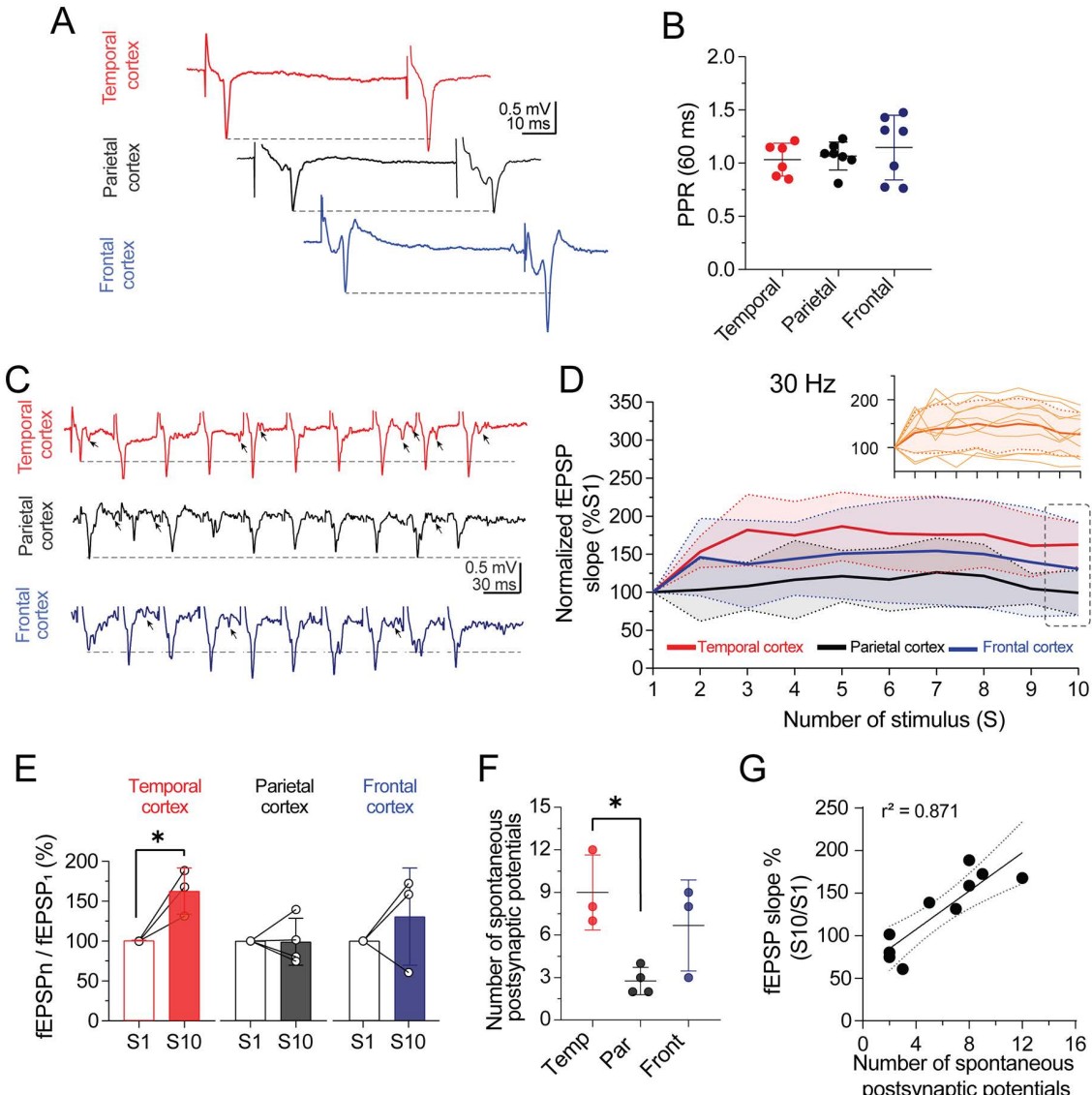

**Fig 4. Layer I/II – V cortical synapses from patients with drug-resistant epilepsy lack frequency-dependent short-term depression. (A)** Representative traces of paired stimulation (60 ms) from the temporal, parietal, and frontal neocortex. **(B)** Dot plot with error bars contrasting the paired-pulse ratio (PPR) between cortical groups. On average, the PPR showed spare facilitation in each cortical synapse. n = 6 slices/3 patients for the temporal slices; n = 7 slices/3 patients for the parietal slices; n = 7 slices/4 patients for frontal slices. **(C)** Representative traces of evoked fEPSPs in response to a train of 10 pulses elicited at 30 Hz in the temporal, parietal, and frontal cortices. The horizontal dashed line represents the facilitation level. Note the appearance of spontaneous postsynaptic potentials during the train stimulation (indicated by arrows). **(D)** Line plot of each evoked synaptic response recorded in Layer V during the stimulation trains delivered in Layer I/II at 30 Hz. At the individual level, the recorded trains exhibited both synaptic depression and facilitation, and the average response was biased toward synaptic facilitation (bold orange line, inset panel D). When sorting the slices by cortex type, temporal synapses showed synaptic facilitation, whereas parietal and frontal synapses did not. **(E)** The bar and line graphs show that only temporal synapses exhibited a significant facilitation level (fEPSP slope % of S10 vs. S1). *p < 0.05, ratio-paired t-test. n = 3 slices/3 patients for temporal and parietal slices; n = 4 slices/3 patients for parietal slices. **(F)** Dot plot with error bars summarizing the number of spontaneous postsynaptic potentials recorded during 30 Hz train stimulation across different neocortical regions. A significantly higher number of events was observed in temporal slices than in parietal slices. *p < 0.05, one-way ANOVA followed by Tukey post hoc test. **(G)** Pearson´s correlation analysis between the number of spontaneous postsynaptic potentials and the magnitude of the tenth synaptic response during 30 Hz train stimulation.

## Morphometric analysis of Layer V pyramidal cells

Because the patch pipettes contained biocytin (see methods), post hoc analyses were used to examine dendritic organization. The analysis was limited to four IT-2 neurons obtained from the parietal and frontal neocortex. The cells exhibited similar morphometric and electrophysiological properties (Fig 5A–B). Seven additional PCs exhibited sparse biocytin labeling, restricted to the somata or random sections of the dendritic ramifications. These cells were discarded.

The axon's initial segment was visually identified for all the reconstructed cells. The axon stemmed from the soma base and had a thin appearance with a mean extension of $54.05 \pm 30.82$ µm (blue traces in Fig 5B). Next, the total number of dendrites and their total length were comparable across cells (Fig 5C; see also Table 2). Likewise, the number and the primary dendritic length exhibited similar values (Figs 5D; Table 2). Furthermore, despite variations in dendritic distribution among cells, their surface areas were similar (projection area for cell 1 = 287.26 µm$^2$; for cell 2 = 336.43 µm$^2$; for cell 3 = 408.71 µm$^2$; for cell 4 = 307.59 µm$^2$). Next, the dendritic tree branching patterns and complexity were evaluated with

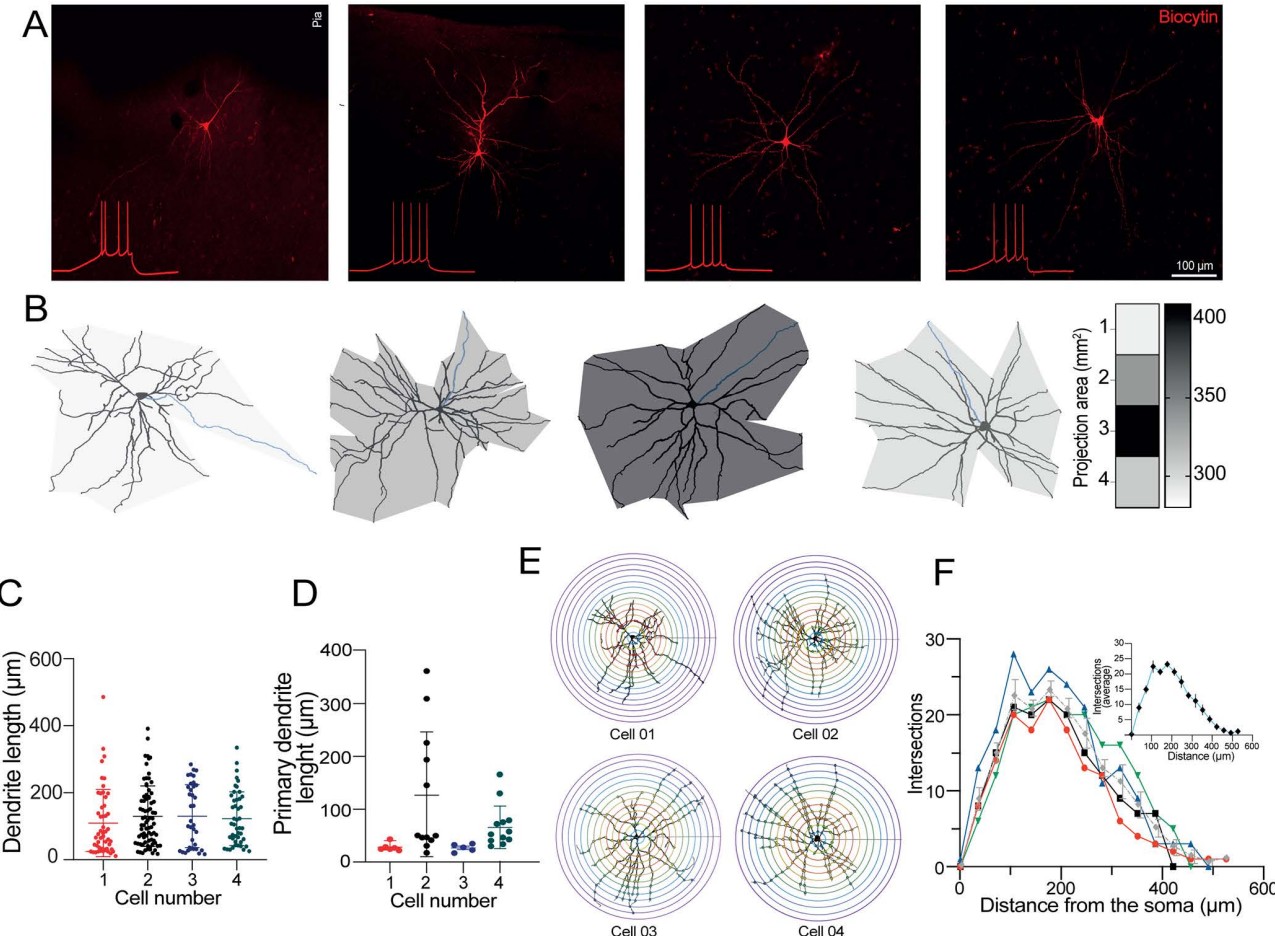

**Fig 5. Morphological analysis of Layer V IT-2 neurons. (A)** Immunofluorescence photomicrographs of PCs filled with biocytin via the patch pipette. The composite images are averages of at least 40 Z-stacks. Inset, firing response of the labeled neurons. **(B)** Projection area of the reconstructed neurons. Dendritic arborizations are shown in black, axon in blue. **(C)** Scatter plot of the total number of dendrites for each cell. **(D)** Scatter plot summarizing the length of the primary dendrite of the four reconstructed cells. **(E)** Sholl projections of the four reconstructed neurons. **(F)** Scatter plot summarizing the number of branch points of each neuron. The inset represents the average from the four reconstructed cells.

**Table 2. Morphological measurements and Sholl Analysis of Layer V pyramidal cells from patients with drug-resistant epilepsy. The dendritic measurements were performed in four RS PCs obtained from the parietal and frontal neocortices.**

**Morphological measurements**

| # Cell | number of total dendrites | total length of the dendrites | number of total primary dendrites | primary dendritic length |
|---|---|---|---|---|
| Cell 01 Parietal | 51 | 107.9±100.7 µm | 6 | 28.35±7.54 µm |
| Cell 02 Parietal | 77 | 129.7±90.26 µm | 14 | 126.6±119.1 µm |
| Cell 03 Frontal | 37 | 130.4±93.64 µm | 6 | 27±7.28 µm |
| Cell 04 Frontal | 46 | 122.7±15.39 µm | 13 | 65.36±38.51 µm |

**Sholl Analysis**

| # Cell | Number of branching points | Branching end tips |
|---|---|---|
| Cell 01 Parietal | 1.49±0.90 | 33 |
| Cell 02 Parietal | 1.41±0.95 | 49 |
| Cell 03 Frontal | 1.40±0.86 | 26 |
| Cell 04 Frontal | 1.10±0.56 | 30 |

a Sholl analysis (Figs 5E and 5F). The quantification of branching points showed no statistical differences between cells (Fig 5F; Table 2), and similar results were observed for branching end tips (Table 2).

## Discussion

This study summarizes a series of intrinsic and synaptic properties of Layer V PCs from three neocortical regions surgically resected from patients with DRE. At the cellular level, we used patch-clamp recordings to classify PCs as IT-2 or ET neurons based on their firing patterns and to determine subthreshold and suprathreshold membrane properties. At the circuit level, extracellular recordings were used to evaluate synaptic strength, coupling between presynaptic volleys and excitatory postsynaptic potentials, and frequency-dependent short-term plasticity within the gamma-range. Significantly, PCs from deep cortical layers, such as Layer V, have been identified as a major source of epileptiform discharges in both animal models and human patients [19,21].

### Electrophysiological differences between IT-2 and ET neurons

In this study, we identified two main neuronal populations in the human neocortex: IT-2 and ET neurons. Although these populations can be reliably distinguished by their spiking patterns [24,37,38]—regular spiking for IT-2 neurons and bursting for ET neurons—we also observed other well-recognized electrophysiological features that differentiate these neuronal types, including spike adaptation, instantaneous firing frequency, and input resistance [24]. Interestingly, previous studies have shown that ET neurons display a larger sag ratio than IT-2 neurons [24], a difference explained by their higher expression of HCN channels [39]. However, in our study, we did not observe differences in sag ratio between IT-2 and ET neurons from patients with DRE, suggesting that a possible downregulation in the functional expression of HCN channels, at least in ET neurons, may explain this finding. In support of this idea, Layer II/III pyramidal neurons from human epileptic neocortex shows a reduced $I_h$ mediated by HCN channels [40].

Another striking finding was the apparently higher input resistance in both IT-2 and ET neurons compared previous works [24], despite the maintained proportional difference between the two populations. While these differences could be attributed to methodological differences such as composition of cutting solution (NMDG-based vs. sucrose-based), as recently discussed [41], the higher input resistance observed in our study may reflect an intrinsic feature related to pathological nature of the tissue examined. For instance, Kalmbach et al. used tissue considered non-pathological by use tissue obtained from distant from epileptic or tumoral zones, we used pathological tissue directly obtained from epileptic foci. In

addition, consistent with the possible functional downregulation of HCN channels mentioned previously, blockade of HCN channels in human cortical neurons has been shown to significantly increase input resistance and hyperpolarize the RMP [42]. Therefore, theoretically, a reduced functional expression of HCN channels may explain the higher input resistance and hyperpolarized RMP observed in our study. While we cannot rule out a possible contribution from dendritic damage during the slicing process to measurements such as input resistance [16], our results, together with previous evidence, point to a potential dysregulation of HCN channels in influencing intrinsic membrane properties under pathological conditions such as epilepsy.

## Membrane properties of IT-2 neurons

Despite extensive knowledge of the neuronal physiology of rodent neurons, our understanding of the human neocortex's intrinsic electrophysiological properties is still limited. Moreover, a recent study showed that intrinsic properties, such as somatic $R_N$, rheobase current to evoke an AP, HCN channel-mediated currents, and PC firing, differ between rodent and human neocortices [16].

Given the inherent challenge of obtaining healthy human brain tissue for comparative analysis [11], in most studies, tissue resected from areas away from epileptic zones or deep brain tumors is regarded as 'normal' or 'moderately less affected' by neuropathologists [16,43]. The neurons examined in our study exhibited hyperpolarized RMP, minimal SSA, and firing outputs resembling those observed in rodent and human neocortices under normal or near-physiological conditions [16,44,45]. However, the homogeneity of the passive property measurements with the patch-clamp recordings suggests that a subset of ion channels active near the RMP of these cells did not experience significant changes, thereby maintaining the ionic distribution across the cellular membrane. Nevertheless, the higher excitability observed in frontal IT-2 neurons compared to parietal and temporal neurons may be explained by differences in the biophysical properties of rapidly activating $Na^+$ channels rather than by changes in passive properties. For instance, while our measurements of MDS, also referred to as the rise speed of the AP [14], suggest a higher global $Na^+$ conductance in frontal neurons, this measurement may also be influenced by biophysical properties of $Na^+$ channels such as voltage dependence, conductance density (i.e., number of channels), maximal conductance [14], and the subunit composition of the channels [46]. Despite these considerations, previous studies in mice and macaques have reported regional differences in the properties of AP and the firing frequency of Layer V pyramidal neurons [39,47]. Therefore, the observed regional differences in excitability may reflect the functional specialization of each cortical region (e.g., executive functions in the frontal cortex or sensory processing in the parietal lobe).

Another important consideration is that electrophysiological measurements may vary across life stages [48]. For instance, the RMP in Layer II/III PCs from non-epileptic tissue becomes more hyperpolarized from childhood to early adulthood (mean: –65 mV vs. –69 mV). In our study, despite age differences between the examined cortices (2–7 years, childhood, for the parietal cortex; 21–30 years, early adulthood, for the temporal and frontal cortices), the RMP was predominantly more hyperpolarized (–72 to –73 mV) than in cortical neurons from non-epileptic tissue [22,48]. These observations support the idea that a more hyperpolarized RMP may be an underlying feature of epileptic tissue. However, it is worth noting that other factors, such as series resistance and liquid junction potential, can also influence RMP measurements.

## Changes in the synaptic transmission and short-term plasticity of the neocortical synapses

Within neocortical synaptic networks, the postsynaptic firing is influenced, among other factors, by the dynamics of presynaptic release and the short-term plasticity of synaptic transmission [36,49]. Consistent with findings in the rodent neocortex, frequency-dependent STD modulates information transfer efficiency among human PCs across neocortical layers under near-physiological conditions [15]. Likewise, STD balances neocortical activity during prolonged synaptic stimulation [36], positing that frequency-dependent STD is a key cellular mechanism for regulating the propagation of electrical

activity across neocortical layers. With that in mind, our extracellular recordings uncovered a lack of synaptic depression in response to gamma-range stimulation of the superficial neocortical Layers I/II, compared to previous observations [15]; even moderate synaptic facilitation was observed in the temporal neocortex (Fig 4C–D). Consequently, the gain of frequency-dependent synaptic facilitation, rather than synaptic depression [15,35], may represent a potential mechanism that favors network hyperexcitability during seizure activity by amplifying neuronal activity (e.g., summation of fEPSPs during high-frequency activity). This possibility arises from the observation that synaptic facilitation during spike activity transiently increases the probability of neurotransmitter release, thereby enhancing postsynaptic excitatory conductances [50].

Moreover, we observed that the frequency-dependent synaptic facilitation was highly correlated with higher spontaneous postsynaptic potentials. The appearance of spontaneous postsynaptic potentials during 30 Hz train stimulation is striking, and it may be interpreted as a consequence of a transient increase in excitatory activity during brief high-frequency stimulation, as previously discussed. For instance, while recurrent spontaneous activity in deep Layer V of the neocortex is well documented [45], it is typically not observed during brief high-frequency stimulation under physiological conditions in rodents [35] or near-physiological conditions in humans [15]. However, given that this recurrent spontaneous activity is locally controlled by a subset of Layer V pyramidal neurons [51], the potential loss of STD in Layer I/II – V synapses, or alternatively, the emergence of synaptic facilitation (as observed in temporal cortex) during 30 Hz train stimulation, might provide the initial excitatory drive required to trigger such spontaneous activity. Although appealing, this hypothesis requires additional investigation to determine whether the lack of STD or the gain of facilitation during brief high-frequency stimulation contributes significantly to the network hyperexcitability observed in drug-resistant epilepsy.

Lastly, a potential explanation for the lack of frequency-dependent STD is decreased expression of HCN1 channels, as inactivation or blockade of these channels increases temporal summation of synaptic events and promotes epileptiform discharges [52,53]. If this is true, promoting the activity of HCN1 channels in certain types of epilepsy (but see [48]) may restore the frequency-dependent STD and, consequently, prevent or reduce the cortical spread of pathological electrical activity during seizures, as these channels preferentially regulate dendritic excitability and epileptogenesis [54,55]. It is noteworthy that, although calcium concentration strongly modulates short-term plasticity, in high-probability release synapses, an increase in calcium enhances STD, whereas a reduction favors facilitation [56–58]. In our experiments, we used 2.5 mM $Ca^{2+}$, which is a close to the range (1.3–2 mM) that has been shown to reliably elicit STD in cortical synapses from non-epileptic human tissue [15,59]. Therefore, it is unlikely that the synaptic facilitation observed in response to brief 30 Hz train stimulation can be attributed to the calcium concentration used. Nevertheless, this observation warrants corroboration by future studies.

In the human cortex, presynaptic release probability has been poorly explored, yet it is documented that Layer II/III synapses in the temporal neocortex from non-epileptic tissue exhibit paired-pulse depression [15]. This is consistent with previous studies using dual-patch-clamp recordings from PCs in the rat neocortex [36]. In this sense, we found that paired stimulation (60 ms) at all neocortical synapses did not trigger depression; contrary to the expected, a mild paired-pulse facilitation was observed. This apparent polarity shift suggests a change in the neurotransmitter release process from high to low release probability [60]. We hypothesize that reduced synaptic depression (i.e., PPF) in the temporal neocortex increases excitability during high-frequency activity, as observed during seizures. A similar phenomenon has been previously documented in murine neurons [61].

## Limitations of the study

The inherent difficulty in obtaining healthy human brain tissue for comparative research is a key limitation of this study. Additional weaknesses include variations in the tissue age, limited analyses of neurotransmitter release probability, and sparse post-hoc reconstructions obtained from recorded cells. Despite these constraints, we emphasize that acute human neocortical slices, combined with whole-cell patch-clamp and extracellular recordings, offer a robust, high-resolution,

cell- and circuit-level approach to understanding the neuronal physiology of human cortical neurons under pathological conditions.

## Conclusion

Our findings suggest that Layer V neocortical IT-2 neurons from patients with drug-resistant epilepsy are not inherently hyperexcitable at the somatic level. Instead, their firing frequency is comparable to that of unaffected human tissue [16]. At the synaptic level, however, alterations in the PPR, decoupling between presynaptic excitability and postsynaptic response, and the lack of frequency-dependent STD may contribute to the hyperexcitable state observed during seizure activity. Future studies should aim to analyze the individual contributions of cellular elements within neuronal compartments to clarify their specific roles in the hyperexcitability underlying seizure activity.

## Supporting information

**S1 File. Detailed information of the recorded neurons.** Individual data included in the study, organized by neuron type and experimental protocol applied. Each Excel tab contains the neurophysiological values processed for each figure in the manuscript.
(XLSX)

## Acknowledgments

We would like to acknowledge R. Olvera for early exploratory experiments.

## Author contributions

**Conceptualization:** Emilio J. Galvan.

**Data curation:** Luis A. Márquez, Isabel Sollozo-Dupont, Emilio J. Galvan.

**Formal analysis:** Luis A. Márquez, Estefanía Gutiérrez-Castañeda, Ernesto Griego, Isabel Sollozo-Dupont, Emilio J. Galvan.

**Funding acquisition:** Emilio J. Galvan.

**Investigation:** Luis A. Márquez, Christopher Martínez-Aguirre, Estefanía Gutiérrez-Castañeda, Emilio J. Galvan.

**Methodology:** Luis A. Márquez, Christopher Martínez-Aguirre, Felix López-Preza, Mario Alonso-Vanegas, Luisa Rocha Arrieta, Emilio J. Galvan.

**Resources:** Luisa Rocha Arrieta, Emilio J. Galvan.

**Validation:** Luis A. Márquez, Isabel Sollozo-Dupont, Emilio J. Galvan.

**Visualization:** Emilio J. Galvan.

**Writing – original draft:** Luis A. Márquez, Emilio J. Galvan.

**Writing – review & editing:** Emilio J. Galvan.

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
