## [Decision Letter · Decision Letter 0]

25 Apr 2025

Dear Dr. Galvan,

Thank you for submitting your manuscript to PLOS ONE. After careful consideration, we feel that it has merit but does not fully meet PLOS ONE’s publication criteria as it currently stands. Therefore, we invite you to submit a revised version of the manuscript that addresses the points raised during the review process.

We look forward to receiving your revised manuscript.

Kind regards,

Michele Giugliano

Academic Editor

PLOS ONE

[Conahcyt to LR A3-S-26782].

Additional Editor Comments (if provided):

Reviewers' comments:

Reviewer's Responses to Questions

**Comments to the Author**

1. Is the manuscript technically sound, and do the data support the conclusions?

Reviewer #1: Partly

Reviewer #2: No

2. Has the statistical analysis been performed appropriately and rigorously?

Reviewer #1: No

Reviewer #2: No

3. Have the authors made all data underlying the findings in their manuscript fully available?

Reviewer #1: No

Reviewer #2: Yes

4. Is the manuscript presented in an intelligible fashion and written in standard English?

Reviewer #1: Yes

Reviewer #2: Yes

Reviewer #1: Marquez et al. examined the membrane and synaptic characteristics of a few pyramidal neurons that were recorded from tissue taken from patients with epilepsy after the suspected seizure origin was surgically removed. According to the results, the primary change that can account for the tissue's hyperexcitability occurs at the synapse level rather than the somatic level.

Given the lack of information on the electrophysiology of neurons from the human brain, especially in diseased situations, the study may be relevant. However, the manuscript requires a significant change before it is ready for publication.

Major:

The location of the tissue of origin must be specified, if it can be found in the surgical data. For instance, whether or not the tissue is hippocampus-derived greatly alters a placement in the temporal lobe. The extent to which the tissue is solely inside the seizure onset zone or includes adjacent tissue that is likely less impacted by the clinical history must also be determined. In this context, the available surgical outcome must also be described in the table of affected individuals. This aids in determining the actual contribution of the excised tissue to each patient's network hyperexcitability.

There are too many details missing to fully comprehend the dataset's nature. For each patient, how many slices were taken? It is necessary to provide further details on the patient from whom the (few) recorded cells originate. For instance, one patient may have had four of the six temporal lobe cells recorded, while the other two patients may have just had one cell.

The statistics are presented in a too careless manner. The authors must try to justify the validity of applying their tests to such small samples for each dataset. The number (n) of measurements from which each reported mean value is derived must be included, particularly when using a repeated measures approach. Even if the result is not significant, it still needs to be reported together with the relevant p-value and other information. To clearly identify which factors were taken into account between-subjects and which were evaluated within-subjects, the mixed-effects ANOVA analysis has to be more thorough.

From what is said in the abstract, the focus on the cell morphology results, which are based on just (!) four cells, must be reduced.

Reviewer #2: 1. Is the manuscript technically sound, and do the data support the conclusions?

The manuscript presents original research on the electrophysiological and morphological properties of human Layer 5 (L5) pyramidal neurons (PCs) from different cortical regions, obtained from surgical resections of drug-resistant epilepsy (DRE) patients. The study addresses a relevant and timely topic in human neurophysiology. However, several technical and interpretative issues need to be addressed:

• Methods: Important experimental details are missing, including the osmolarity of all solutions and the rationale for using an unusual transport solution (320 mM sucrose, 1 mM EDTA, 5 mM Tris-HCl), which could affect neuronal viability and physiology due for example to a poor ion balance given by the absence of most of the ions needed to mimic the CSF and so maintain physiological conditions.

The unusually low biocytin concentration (0.04%) is concerning, as it may limit dendritic and axonal filling. The methods section does not specify the number of cells analyzed for dendritic complexity, nor the criteria used for their selection. However, based on the results, it appears that only four cells were included in this analysis. This raises concerns regarding the representativeness and robustness of the findings, as such a small sample size limits the ability to draw meaningful conclusions. Similarly, the small sample size used for the classification of L5 PCs across the different cortices makes generalization difficult.

• Data quantification: Several claims are not adequately supported by quantified data or statistical tests (e.g. lines 287, 296, 300-311). Notably, parameters such as action potential (AP) firing rates, adaptation, and afterhyperpolarization are mentioned but not systematically quantified or statistically compared.

• Neuron classification: The classification of L5 PCs based solely on spike frequency adaptation and firing rate is oversimplified. Extensive literature demonstrates the existence of well-defined subpopulations (ET vs. IT; for human temporal cortex gyrus see for example Kalmbach et al., 2021; for rodent neocortex see Moberg and Takahashi, 2022). These cells have distinct morphological and electrophysiological features that must be taken into account while attempting to classify L5 PCs in human cortex. The authors should revisit their classification strategy in light of established frameworks.

• Results interpretation: The interpretation of the FV-fEPSP relationship and the claim of synaptic imbalance in DRE lacks proper statistical comparison (e.g., regression slopes) and validation against healthy tissue or additional rodent data. Several passages are speculative and need either experimental evidence (e.g., recordings with synaptic blockers) or should be moved to the discussion.

Additionally, the morphological analysis in this study raises concerns about the representativeness and robustness of the findings, as the small sample size and large neuronal heterogeneity in human L5 (e.g., Kalbach et al, 2021) limit meaningful conclusions. Reported values, such as a total dendritic length of ~100 μm and an average of 1.5 branches, are unusually low compared to published data (e.g., >10 mm dendritic length and >60 apical branches in Mohan et al., 2015; Kalbach et al., 2021). These discrepancies may result from technical issues, including low biocytin concentration (0.04%?) or tissue damage during slicing. While the authors briefly mention some limitations, they still draw conclusions as if these morphological parameters were fully representative. Given the small sample size, possible artifacts, and inconsistencies with previous literature, these findings should be interpreted with caution, and further validation with larger, methodologically optimized datasets is strongly recommended.

• Figures: Several figures lack error bars, sample size annotations, and statistical comparisons. Some traces are presented without supporting quantification and should be either properly quantified or removed. Additionally, certain visualizations are unnecessarily complicated. For example, in Fig. 1B, expressing the number of cells as percentages makes the information less accessible than it needs to be, and the actual numbers should be reported directly. In Fig. 5C, D, and F, the heatmaps could be replaced with simple numeric values, as only four cells were analyzed.

In summary, while this study explores an important and promising area, substantial revisions are required to ensure that the conclusions are rigorously and appropriately supported by complete, quantified, and statistically tested data.

2. Has the statistical analysis been performed appropriately and rigorously?

Statistical analyses are generally described but are not always rigorously applied:

• Missing analyses: Many important comparisons (e.g., between cortical regions, between firing pattern types) lack proper statistical testing or explicit reporting of sample sizes. For example, in lines 430-438, the description of spontaneous synaptic events following train stimulation requires clarification. It is unclear whether these events represent recurrent excitation, asynchronous release, or another process. The claim that the number of these events “tended to be higher” in the temporal cortex lacks statistical validation. Further analysis is needed to determine whether this facilitation reflects pathological hyperexcitability or normal synaptic dynamics.

• Choice of error metrics: The use of the standard error of the mean (SEM) rather than standard deviation (SD) with such small sample sizes can give a misleading impression of data variability. SD would be more appropriate to accurately reflect the dispersion within the sample, particularly in cases where individual variability is of interest.

• Linear regression analyses: The interpretation of r² values in the FV-fEPSP relationship needs to be more cautious (lines 399-406). Perfect or near-perfect r² values (such as an r² = 1 in the frontal cortex data) may indicate overfitting that might be given by some issues in the data such as limited data points. Additionally, the phrase “theoretical consistency and proportionality” is vague and should be explicitly defined how the FV-fEPSP relationship in DRE in altered by comparing it with “healthy” tissue (i.e. non-epileptic, for example access tissue from tumor resections) or more rodent data. The least squares method quantifies the strenght of a linear relationship, but it does not inherently substantiate or prove synaptic imbalance. A statistical comparison of regression slopes across conditions is needed to support this claim. Finally, the claim that the proportionality is "disrupted" in DRE should be backed by statistical evidence.

• Assumptions and justifications: The rationale behind choosing specific current injection ranges and classification criteria is unclear and requires clarification. For example, was the range 90-210 pA empirically chosen, or was the firing linear without any adaptation, or was there the absence of significant variability in firing rates?

3. Have the authors made all data underlying the findings in their manuscript fully available?

The data availability is generally acceptable but lacks clarity in several areas:

• The number of cells analyzed for dendritic complexity is not specified in the methods, though results suggest only four were used, raising concerns about representativeness.

• The availability of raw electrophysiological data (e.g., individual firing frequencies, membrane properties, and fEPSP measurements) is not explicitly confirmed.

• Data supporting several figure panels (e.g., spontaneous postsynaptic potentials, firing rates) are either missing or insufficiently described.

The authors should ensure that all underlying data points (not only summary values) are made available.

4. Is the manuscript presented in an intelligible fashion and written in standard English?

The manuscript is generally intelligible and written in acceptable English, but several issues of clarity, consistency, and terminology need to be addressed:

• Abbreviation consistency: All abbreviations must be defined at first appearance and used consistently throughout the text.

• Sentence clarity: Several sentences in the Introduction and Methods are overly long or ambiguous (e.g., lines 69-73 and 74-79). These should be split and clarified.

• Technical terminology: Terms like “synaptic spontaneous activity” should be replaced by the more precise “spontaneous postsynaptic potentials.” In line 162, it is not clear what the authors mean with “interval’s latency”. If I understand Fig. 1 correctly, the latency refers to the time between the onset of the current injection and the occurrence of each evoked action potential within that step - is this what the authors meant? It should be also better clarified whether the authors refer to cytoarchitectonic subdivisions of Layer 5 (e.g., 5a vs. 5b) or a particular distance from the pia when they write “the middle region of layer V” (line 277) or “external region of layer V” (line 375).

• Consistency in tense and form: Active/passive forms should be harmonized, and descriptions of results made more precise.

• Figure labels: Some figure labels are misleading or incorrect (e.g., calling voltage traces “I-V curves” in Fig. 2B1-B3).

Overall, while the manuscript is understandable, careful editing for clarity, grammar, and terminology is needed.

Summary:

Major points

• Please add missing details in the Methods section, including osmolarity of solutions, rationale for transport medium, objective lenses, z-stack step size, and sample size criteria for morphology analysis.

• Reconsider the classification strategy for Layer 5 pyramidal neurons in light of established subtypes (ET/IT) as described in recent literature (e.g. Kalmbach et al., 2021; Moberg and Takahashi, 2022).

• Quantify all electrophysiological parameters discussed in the text (e.g., AP firing rates, spike adaptation, afterhyperpolarization), and report statistical analyses.

• Correct overinterpretations regarding synaptic influence on firing rates and the FV-fEPSP relationship. These claims should be either experimentally tested (e.g., with synaptic blockers) or restricted to the discussion.

• Review and improve figure presentations, i.e. add error bars, report the number of cells, perform statistical comparisons, and remove unsupported traces.

• Replace speculative interpretations with data-supported statements or move them to the discussion.

• Revise the language for clarity, consistency, and technical accuracy, addressing long or ambiguous sentences, terminology, and typographical errors.

Minor points

• Adjust terminology, e.g. “spontaneous postsynaptic potentials” instead of “synaptic spontaneous activity.”

• Clarify specific method descriptions (e.g., “Membrane capacitance was calculated as the ratio of the membrane time constant to the input resistance”).

• Correct typographical and figure annotation errors (e.g., r² values in line 398, significance asterisks in Fig. 3C).

**Do you want your identity to be public for this peer review?** For information about this choice, including consent withdrawal, please see our Privacy Policy

Reviewer #1: No

Reviewer #2: **Yes:** Laura Monni

---

## [Author Response · Author response to Decision Letter 1]

2 Jul 2025

Here is the list of changes according to the comments raised by the reviewers:

Reviewer #1

Marquez et al. examined the membrane and synaptic characteristics of a few pyramidal neurons that were recorded from tissue taken from patients with epilepsy after the suspected seizure origin was surgically removed. According to the results, the primary change that can account for the tissue’s hyperexcitability occurs at the synapse level rather than the somatic level.

Given the lack of information on the electrophysiology of neurons from the human brain, especially in diseased situations, the study may be relevant. However, the manuscript requires a significant change before it is ready for publication.

Major:

The location of the tissue of origin must be specified, if it can be found in the surgical data. For instance, whether the tissue is hippocampus-derived greatly alters a placement in the temporal lobe. The extent to which the tissue is solely inside the seizure onset zone or includes adjacent tissue that is likely less impacted by the clinical history must also be determined. In this context, the available surgical outcome must also be described in the table of affected individuals. This aids in determining the actual contribution of the excised tissue to each patient’s network hyperexcitability.

First, we would like to acknowledge the time and effort made by the reviewer in reading our manuscript. We have taken into consideration all his/her comments and we have integrated them in both this response letter and the new version of the manuscript.

We understand the relevance of the tissue’s origin and how this factor may influence the electrophysiological properties of the recorded neurons. The cortical tissue included in our study was obtained from patients diagnosed with drug-resistant epilepsy who underwent surgery to resect the brain tissue identified as the epileptic focus. First, the epileptic foci were determined using intraoperative electrocorticography. This gold-standard method identifies epileptiform activity and guides the extent of pathological tissue to be resected. The tissue used in our electrophysiological recordings was derived from these resections, and we assume that most of our samples originated directly from the epileptic focus.

We are sorry to say that the exact anatomical location of each tissue sample is unknown, as the resected material was subsequently sectioned for different studies, including histopathological analyses, biochemical measurements, and electrophysiological analyses. Due to the loss of precise localization of the tissue used for the electrophysiological recordings, we present the results by cortical area type. On the other hand, based on the intraoperative ECoG findings by the surgical team, we understand that most of the resected tissue was pathological. For obvious ethical reasons, it is not possible to obtain healthy, non-pathological tissue for comparative purposes.

Regarding the temporal lobe samples, while most cases of drug-resistant temporal lobe epilepsy affect the hippocampus, the samples in this study were obtained from non-hippocampal regions of the temporal lobe. Samples of that nature were collected and analyzed for different studies that are beyond the scope of this study. It is, therefore, likely that the epileptic foci in these cases were in other areas, such as the entorhinal or post-rhinal cortices.

Nonetheless, in the updated version of our manuscript, we have clarified in the methods section that the examined tissue was obtained from epileptic foci identified by intraoperative electrocorticography, an experimental technique that confirms their hyperexcitable state at the network level. We hope that the reviewer understands that we cannot provide details regarding the anatomical location of the brain tissue, as the resected tissue was divided in pathological studies for clinical purposes and experimental samples for biochemical and electrophysiological experiments. Despite this limitation, we believe that the quality of the electrophysiological experiments and post-hoc reconstructions provides solid experimental evidence of the activity of human neurons obtained from brain with drug-resistant epilepsy.

Following the reviewer’s comment, the amendment included in the upgraded version of the manuscript reads on page 6 as follows: The resected brain samples were obtained directly from epileptic foci identified by intraoperative electrocorticograms, which confirm their hyperexcitable nature at the network level.

There are too many details missing to fully comprehend the dataset’s nature. For each patient, how many slices were taken? It is necessary to provide further details on the patient from whom the (few) recorded cells originate. For instance, one patient may have had four of the six temporal lobe cells recorded, while the other two patients may have just had one cell.

Before addressing the reviewer’s comment, we would like to emphasize that the data presented in this study are derived from whole cell patch-clamp experiments performed on human tissue obtained directly from surgical procedures. In such cases, it is not possible to establish or preselect the number of cells recorded per patient or to standardize the number of cells to be included in the study. One of the main motivations for submitting this study was the exceptional quality of the electrophysiological recordings. We are aware of the comment that we have a limited number of cells, but the quality of the recordings should be taken into account, rather than focusing solely on the limited number of cells. These are human neurons, not rodent ones. Statistical analyses serve to validate or reject data; we conducted a rigorous analysis for each parameter to validate our findings. With that said, we acknowledge the reviewer’s concern, and we have addressed this issue by including a supplementary database. This database contains all raw electrophysiological measurements corresponding to each Figure, along with detailed information on the number of recorded cells or slices per patient and by cortical region. In the updated manuscript, we present the sample size as n = number of cells / number of samples from 10 patients. For instance, IB neurons, now referred to as extratelencephalic (ET) neurons, have a sample size of n = 10 cells / 4 samples from 10 patients. This indicates that 10 ET neurons were found in only 4 out of the 10 patients. The specific origin of each cell or slice is provided in the supplementary dataset.

The statistics are presented in a too careless manner. The authors must try to justify the validity of applying their tests to such small samples for each dataset. The number (n) of measurements from which each reported mean value is derived must be included, particularly when using a repeated measures approach. Even if the result is not significant, it still needs to be reported together with the relevant p-value and other information. To clearly identify which factors were taken into account between-subjects and which were evaluated within-subjects, the mixed-effects ANOVA analysis has to be more thorough.

The use of parametric statistical tests was based on the assumption of normal distribution, which was confirmed through normality testing. Although we previously stated that normality was assessed using Kolmogorov-Smirnov tests, which were used in most cases, we also performed Shapiro-Wilk tests (originally unaddressed in the methods section). Unlike other normality tests, such as D’Agostino-Pearson or Kolmogorov-Smirnov tests, the Shapiro-Wilk test is specifically suited for small sample sizes and can be used with as few as three observations. In other words, the Shapiro-Wilk test is more sensitive for detecting deviations from normality. In the revised version of the manuscript, we now clarify that the normality of the data was validated using a Shapiro-Wilk test, which is appropriate for small sample sizes. This amendment is included in the methods section where it reads: The normality distribution of data was validated and assessed using the Shapiro-Wilk test, which is appropriate for small sample sizes with a minimum of three observations (n = ≥3).

Regarding the number “n” of measurements, we have revised the manuscript and we have added the number (n) of sample sizes for each electrophysiological parameter examined. Likewise, we selected a mixed-effects ANOVA because our design included two relevant factors influencing firing rate: the cortex type (a between-subjects factor) and current injection (or within-subject factor), following the reviewer’s comment. It is important to note that multiple current levels (0 to 390 pA, 30 pA steps) were somatically injected into the cell, resulting in repeated measures. Given that in two recorded cells, the firing rate values at higher current levels (e.g, 390 pA) were missing, we used mixed-effects ANOVA. This test is well suited to handle unbalanced data and missing values in repeated-measures designs and to account for within-cell variability by including random effects. While it is expected that increasing current magnitude leads to higher firing rates, our main aim was to determine whether firing rate is significantly influenced by cortex type. Nevertheless, following the reviewer´s observation, we have now clarified in the data analysis section that cortex type and current injections correspond to between-subjects factor and within-subject factor, respectively. The revised sentence now reads as follows:

In case of mixed-effects ANOVA, we analyzed cortex type and current injection factors as between-subjects factor and within-subject factor, respectively.

We hope this clarification helps the reviewer understand our rationale for not including the current injection factor (within-subject factor) as a primary explanatory variable for firing rate.

From what is said in the abstract, the focus on the cell morphology results, which are based on just (!) four cells, must be reduced.

Although we believe that including these neuronal reconstructions and morphometric data was relevant due to the information they provided, we also acknowledge the limitations of our dataset, as each human neuron was electrophysiologically characterized at the single-cell level using the whole-cell patch-clamp technique. It is worth noting that we performed biocytin labeling in every cell recorded for this and other studies (from the same tissue sample), and the results were, unfortunately, inconsistent. In some cases, we successfully stained axons, somata, and dendritic fields. However, in a larger part of the post-hoc manipulations, we obtained strong somatic labeling but no dendritic fluorescence or sparse dendritic fluorescence without somatic integrity. It is not easy to contain the excitement when a human neuron, from which all the electrophysiological data were obtained, also exhibited adequate labeling for dendritic reconstructions extending over 400 μm and a consistent distribution of dendritic bifurcations. We believe that this is not a minor or everyday achievement in electrophysiology, or even more importantly, in human electrophysiology using brain samples obtained from the operating room. However, we also acknowledge that both reviewers noted the limitations of this attainment. For this reason, we temper the description and minimize its presentation in the results section. In the upgraded version of the manuscript, we limited our statements to mentioning that a subset of pyramidal cells, which were electrophysiologically characterized, were also morphologically identified.

Reviewer #2:

1. Is the manuscript technically sound, and do the data support the conclusions?

The manuscript presents original research on the electrophysiological and morphological properties of human Layer 5 (L5) pyramidal neurons (PCs) from different cortical regions, obtained from surgical resections of drug-resistant epilepsy (DRE) patients. The study addresses a relevant and timely topic in human neurophysiology. However, several technical and interpretative issues need to be addressed:

• Methods: Important experimental details are missing, including the osmolarity of all solutions and the rationale for using an unusual transport solution (320 mM sucrose, 1 mM EDTA, 5 mM Tris-HCl), which could affect neuronal viability and physiology due for example to a poor ion balance given by the absence of most of the ions needed to mimic the CSF and so maintain physiological conditions.

First, we would like to acknowledge and thank the reviewer for all his/her valuable comments. Each point raised was carefully considered and thoroughly discussed, and we have made a series of amendments following the constructive criticism. We hope that the revised version of the manuscript meets the expectations outlined by the reviewer.

There is a strong rationale and backup literature to justify the sucrose-based solution to maintain the brain tissue in healthy conditions after its isolation. By using 320 mM sucrose, the osmotic pressure exerted by extracellular sodium is maintained, preventing tissue damage due to the increased excitotoxicity triggered by action potential generation or propagation. This manipulation also decreases Ca2+ influx and the subsequent process of neurotransmitter release; a simple method to reduce neuronal excitability. On the other hand, EDTA serves as a Ca2+ chelator and, indeed, a non-specific inactivator of all types of Ca2+-dependent enzymes that may contribute to apoptosis and cell death following the multiple traumatic events experienced by the tissue: surgery, extraction, mechanical manipulation, and then, slicing. Lastly, Tris-HCl adjusted at an adequate pH works as a buffer that resists pH changes within the biological range and preserves the tissue for experimentation. The sucrose-based solution has been used in neurophysiology for over 70 years, as first reported by Gray and Whittaker (Journal of Anatomy, 1962 Jan; 96(Pt 1):79-88) for the isolation of nerve endings. Over the years, mild modifications have been incorporated into this solution to optimize the acute brain slice process. Current literature is packed with examples of those modifications in which the current sucrose concentration, pH control, and calcium chelation are always present.

Now, regarding the possibility of abnormal neuronal physiology due to this solution, it is an unlikely scenario, as neurophysiology has been performed in acute slices obtained with sucrose-based solutions for at least 50 years, and milestone advances, including multiple forms of synaptic plasticity and intrinsic excitability, have been characterized in slices obtained in sucrose-based cutting solutions. In line with this comment, the visual quality of the electrophysiological findings included in this study does not show signs of low-quality recordings, damaged cells, or any suspicious activity of a dying neuron. By visually inspecting the firing discharge and the amplitude of the action potentials obtained during the prolonged depolarization applied in our experiments, it is noticeable that the cell’s physiological response is in good shape.

Therefore, we used a sucrose solution to preserve the human tissue and minimize damage during transportation from the operating room to the laboratory. After slicing the human tissue, the slice’s recovery was performed in a modified ACSF (details in the Methods) that included physiological concentrations of NaCl, KCl, and CaCl2 to facilitate physiological activation. Furthermore, the experiments were performed in standard ACSF maintained at near-physiological temperatures and with constant exposure to carbogen. This strategy, together with the quality of the electrophysiological responses obtained in this and prior (Biomedicines. 2023 Dec 7;11(12):3237.) or ongoing studies under review (Frontiers in Pharmacology, 2025; ID:1627465.), supports the notion that our experiments capture neuronal activity that closely approximates the physiological activation of human neurons.

The unusually low biocytin concentration (0.04%) is concerning, as it may limit dendritic and axonal filling. The methods section does not specify the number of cells analyzed for dendritic complexity, nor the criteri

---

## [Decision Letter · Decision Letter 1]

4 Oct 2025

Dear Dr. Galvan,

Thank you for submitting your manuscript to PLOS ONE. After careful consideration, we feel that it has merit but does not fully meet PLOS ONE’s publication criteria as it currently stands. Therefore, we invite you to submit a revised version of the manuscript that addresses the points raised during the review process.

We look forward to receiving your revised manuscript.

Kind regards,

Michele Giugliano

Academic Editor

PLOS ONE

**Journal Requirements:**

Reviewers' comments:

Reviewer's Responses to Questions

**Comments to the Author**

Reviewer #2: (No Response)

Reviewer #3: (No Response)

2. Is the manuscript technically sound, and do the data support the conclusions?

Reviewer #2: Partly

Reviewer #3: Partly

3. Has the statistical analysis been performed appropriately and rigorously?

Reviewer #2: Yes

Reviewer #3: Yes

4. Have the authors made all data underlying the findings in their manuscript fully available?

Reviewer #2: Yes

Reviewer #3: Yes

5. Is the manuscript presented in an intelligible fashion and written in standard English?

Reviewer #2: Yes

Reviewer #3: Yes

**Reviewer #2:**  I thank the authors for the thorough and thoughtful revisions to their manuscript. The new version is much improved in clarity and presentation, and several of the concerns I raised in my initial review have been well addressed. I appreciate the substantial improvements made in the revised manuscript, particularly the more careful interpretation of the data. This is a welcome change. While I fully understand the challenges involved in working with human cortical tissue - and I acknowledge the value of such data - I would still encourage the authors to frame their conclusions with appropriate caution given the sample size. The difficulty of the preparation is not in question, but ensuring careful interpretation remains important for the strength of the study.

Overall, I believe the manuscript is close to being suitable for publication pending minor revisions.

I do have a few additional comments and suggestions:

1) Thank you for adding the osmolarity value of the intracellular solution (315–325 mOsm/l). While this value is somewhat high compared to typical physiological intracellular osmolarities (~280–295 mOsm), I understand it may not have impacted the current dataset substantially. However, please report the osmolarity also of the other solutions used for transport, slicing, recovery and recording (extracellular aCSF) for completeness.

2) Please include the quality criteria for inclusion of patch clamp recordings, including series resistance and the max % change of it at the end of the recording. Please also add the calculation of the liquid junction potential and if the RMP was corrected for it.

3) Please specify in the Methods section the membrane potential at which cells were held during current-clamp recordings. Additionally, indicate this value (i.e., the RMP) in the example traces in Figure 1D, both for the responses to current injections and for the spontaneous synaptic activity traces.

4) Line 423: the manuscript states that data from " a control rodent´s acute slice " were used for comparative purposes. However, it appears that the rodent dataset consists of only one slope value (n = 1). If so, this does not support statistical testing, as no variance can be estimated from a single observation - even if a standard deviation (SD = 5.68) is later reported. Unless multiple independent rodent data points are available, I recommend presenting this comparison descriptively, without inferential statistics. If instead this was a typo, please correct it and include also the sample size of rodent’s data in the figure legend.

Moreover, since the data were taken from a publication, please cite it also in the results section.

5) Line 427: minor typo in the value of r2.

6) Please review again the use of abbreviations for consistency. For example, the abbreviation “RS” (regular spiking) is used in the text but is not defined at first appearance (e.g., line 167). While it’s acceptable to remind the reader that IT-2 neurons are regular spiking, for clarity and consistency, please use either “IT-2 neurons” or “RS neurons/PCs” throughout. For instance, in the morphological analysis section, the term “RS PCs” appears, which may be confusing without consistent terminology.

7) Discussion: the authors compare RMP values across studies and relate it to the age limitation (range 2–30 years). However, such comparisons are inherently limited: RMP measurements in whole-cell patch clamp are strongly influenced by recording conditions, including pipette and bath solutions, series resistance, dialysis effects, and the often-uncompensated liquid junction potential. Without standardized protocols, direct comparisons of absolute RMP values between studies are not reliable. Furthermore, the cited age range may not represent a major limitation. Barzó et al. (2025, eLife) analyzed a large dataset of human L2/3 pyramidal neurons (from cortical tissue close to pathological lesion – but in line also with non-human primate and rodent studies) across the lifespan and showed that the most pronounced changes in passive properties, including RMP, occur early in life (before age 1), with relatively stable values beyond that. This suggests that developmental shifts in RMP are unlikely to confound comparisons within the 2–30-year range, provided the data were recorded consistently.

8) Please, be consistent also with citation style as sometimes (Last name et al, year) is provided instead of numbers.

**Reviewer #3:**  Review of “Layer V Neocortical Neurons From Individuals With Drug-Resistant Epilepsy Show Multiple Synaptic Alterations but Lack Somatic Hyperexcitability”

The authors performed whole-cell recordings from human layer-V pyramidal cells resected from the temporal, parietal, and frontal cortex in patients with drug-resistant epilepsy. Cells were classified as IT-2 (regular-spiking) and ET (intrinsic-bursting), with regional analyses primarily focused on IT-2 neurons. Using L I/II stimulation while recording in L Va, they measured presynaptic fiber volleys (FV) and field EPSPs. FV–fEPSP correlation was weak in human tissue but much stronger in a rodent control. A key finding is that 10-pulse, 30-Hz trains produced significant facilitation in the temporal cortex, whereas the frontal cortex showed only mild facilitation, and the parietal cortex showed none. These synaptic results were accompanied by more spontaneous postsynaptic events in the temporal cortex than in the other regions. In conclusion, the authors state that differences in synaptic properties, rather than cellular electrophysiology, underlie epilepsy.

The authors applied solid, classical methods and clearly state their aims. It is also important to enhance the human electrophysiological recording dataset, as done by the authors, rather than relying on a single dominant source of data. Nevertheless, several concerns arose during reading, some about the results themselves and others about the strength and framing of the conclusions. In some cases, the lack of proper control does not prevent publication but requires the authors to adjust their conclusion.

Major comments

------------------

• SSA and time from resection (Figure 4G).

SSA appears highly related to elapsed time after surgery/slicing (larger immediately post-resection). Did the authors record the time from surgery to experiment for each neuron and analyze SSA versus this interval?

• Sag ratio (IT-2 vs ET).

Reported sag ratios (IT-2: 0.141 ± 0.072; ET: 0.164 ± 0.063) are smaller than expected from prior work and my own experience. Since rodents were also recorded, a rodent sag comparison could be informative. As it stands, the small difference raises concern that the ET vs IT-2 classification may be compromised.

• Input resistance of ET neurons.

ET Rin of 122 ± 24.1 MΩ seems high relative to prior studies (around 50 MΩ). This calls into question the classification and could impact conclusions about IT-2 properties. Please reconcile with the literature and methods.

• Rheobase.

Figures 2F and 2I appear inconsistent regarding rheobase. In Fig. 2I, the fitted curves suggest the smallest rheobase in parietal, not frontal, cortex. Fits in Fig. 2I should correspond to measured rheobases in Fig. 2F; please reconcile this mismatch, as it affects a main conclusion.

• “Loss of STD”.

The statement “These data suggest that frequency-dependent synaptic facilitation or the loss of STD in epileptic neocortex favors excitatory synaptic activity.” is not supportable without a proper control to link these changes to disease. Therefore, the conclusion should be reframed to avoid implying pathological loss without a proper control. Nevertheless, as I understand the difficulty in getting control tissue, therefore, it is important to underline that the results themselves are valuable.

• Morphology sample size.

Only four biocytin-filled RS PCs (from parietal and frontal cortex) were analyzed. Given known variability with depth, cortical folding, and location within a slice, even within a small region, this is insufficient for strong conclusions. The authors already considered this issue for reviewers #1 and #2, but unfortunately, the issue still stands. Namely, the similarity between the cells could be either statistically significant or not.

Important points

• External Ca²⁺.

Because [Ca²⁺]o critically sets release probability, it can strongly alter short-term plasticity. Please, either provide a control demonstrating that your [Ca²⁺]o choice does not change the qualitative outcome, or discuss how different [Ca²⁺]o would impact interpretation.

• “Barely exhibited SSA at RMP.”

If PCs “barely exhibited SSA at their RMP,” what are the consequences for analyses that use SSA as a readout? Please clarify how SSA was ultimately used in Figure 4G.

• Rheobase vs threshold.

If rheobase differs across regions while RMP and Rin are similar, this implies a lower voltage threshold for AP initiation in the frontal cortex (≈ 60% closer to rest than the temporal). Can this be confirmed directly from existing data?

• “Higher global Na⁺ conductance in frontal PCs.”

Could differences in Na⁺ channel kinetics or subtype expression also explain the observations, beyond “global conductance”? A more nuanced report on the observation would help.

• AP dV/dt variance.

Reported slopes (e.g., −83.9 ± 37.1 mV/ms temporal; −48.2 ± 1.96 mV/ms parietal; −77.5 ± 27.4 mV/ms frontal) show a strikingly smaller SD in parietal neurons. Please explain this disparity.

• Human vs rodent slope comparison.

The statement that slopes differ significantly from the rodent control and therefore indicate “altered synaptic response dynamics” is hard to interpret. Why would human microcircuits be expected to match rodent slopes, given anatomical differences (layers and size proportions of the different layers)? Please justify or reframe.

• Use of predicted values.

In figure 4, the authors plot the FV-fEPSP predicted values (from the fitted line). I have to admit that understanding the reasoning for this plotting is not easy. Please consider a clarification for this panel not to burden the reader

• AIS identification.

“The axon’s initial segment was visually identified.” How was the end of the AIS determined? Please reference standard criteria for AIS identification.

Minor points

--------------

• The Introduction mentions “different neuronal mechanisms at different organizational levels.” Please define these levels explicitly.

• “Glutamatergic strength, the synchronicity between presynaptic volleys and field EPSPs, and short-term, frequency-dependent plasticity were determined at the synaptic level.” Please clarify this phrasing; synaptic properties are, by definition, at the synaptic level.

• “To explore a possible desynchronization… FV vs fEPSP slopes were correlated.” How is “desynchronization” defined here? Do you mean reduced coupling or statistical independence between the two signals?

• “≈ 65% of the recorded neurons belong to the IT-2 neurons…” Consider: “≈ 65% of recorded neurons belong to the IT-2 class/ were IT-2.”

• “Extracellular recordings in independent slices.” Please define “independent” (e.g., different patients, different blocks, or non-overlapping sites?).

• “Strong linear correlation and slope recovery… despite disorganized or desynchronized conditions.” The terms “disorganized” and “desynchronized” are not clear in this context. Please clarify, define precisely, or revise.

**Do you want your identity to be public for this peer review?** For information about this choice, including consent withdrawal, please see our Privacy Policy

Reviewer #2: **Yes:** Laura Monni

Reviewer #3: No

---

## [Author Response · Author response to Decision Letter 2]

21 Nov 2025

Responses for reviewers’ comments

Review 2

I thank the authors for the thorough and thoughtful revisions to their manuscript. The new version is much improved in clarity and presentation, and several of the concerns I raised in my initial review have been well addressed. I appreciate the substantial improvements made in the revised manuscript, particularly the more careful interpretation of the data. This is a welcome change. While I fully understand the challenges involved in working with human cortical tissue – and I acknowledge the value of such data - I would still encourage the authors to frame their conclusions with appropriate caution given the sample size. The difficulty of the preparation is not in question, but ensuring careful interpretation remains important for the strength of the study.

Overall, I believe the manuscript is close to being suitable for publication pending minor revisions. I do have a few additional comments and suggestions:

1) Thank you for adding the osmolarity value of the intracellular solution (315–325 mOsm/l). While this value is somewhat high compared to typical physiological intracellular osmolarities (~280–295 mOsm), I understand it may not have impacted the current dataset substantially. However, please report the osmolarity also of the other solutions used for transport, slicing, recovery and recording (extracellular aCSF) for completeness.

R: We have added this information to the revised manuscript. The osmolarities of the solutions were approximately 315 mOsm for the transport solution, 290 mOsm for the slicing solution, 315–320 mOsm for the incubation solution, and 290–295 mOsm for the aCSF solution. These values are now shown in methods section.

Also, we followed the reviewer comments, and we included a “limitations of the study” paragraph in the discussion section of the study. We have temper our interpretations and acknowledge our limitations in the study.

2) Please include the quality criteria for inclusion of patch clamp recordings, including series resistance and the max % change of it at the end of the recording. Please also add the calculation of the liquid junction potential and if the RMP was corrected for it.

R: The series resistance values recorded ranged from 14 to 25 MΩ, and cells that exhibited a >20 % change were excluded from analysis. The liquid junction potential was 12–15mV, and the RMP values were not corrected for it. This information was added to the methods section (lines 199 to 201).

3) Please specify in the Methods section the membrane potential at which cells were held during current-clamp recordings. Additionally, indicate this value (i.e., the RMP) in the example traces in Figure 1D, both for the responses to current injections and for the spontaneous synaptic activity traces.

R: For current-clamp recordings, intrinsic membrane properties were determined at RMP, and the firing measurements were performed at a holding potential of −70 mV. This information is included in the Methods section of the revised version of the manuscript. Since representative traces by firing type and SSA traces were obtained from the same cell, we have added the RMP value on the left side of each representative trace by firing type in Figure 1D.

4) Line 423: the manuscript states that data from “a control rodent´s acute slice” were used for comparative purposes. However, it appears that the rodent dataset consists of only one slope value (n =1). If so, this does not support statistical testing, as no variance can be estimated from a single observation - even if a standard deviation (SD =5.68) is later reported. Unless multiple independent rodent data points are available, I recommend presenting this comparison descriptively, without inferential statistics. If instead this was a typo, please correct it and include also the sample size of rodent’s data in the figure legend. Moreover, since the data were taken from a publication, please cite it also in the results section.

R: Thanks for noticing this mistake. It was a typo, now corrected to “control rodent’s acute slices.” For the correlation analysis in the rodent cortex, we used measurements of 91 events (fEPSPs) obtained from 5 slices across 3 animals, which are now included in the revised version of our manuscript both results section and corresponding figure legend.

5) Line 427: minor typo in the value of r2.

R: Thanks for this observation. We had corrected the mistake.

6) Please review again the use of abbreviations for consistency. For example, the abbreviation “RS” (regular spiking) is used in the text but is not defined at first appearance (e.g., line 167). While it’s acceptable to remind the reader that IT-2 neurons are regular spiking, for clarity and consistency, please use either “IT-2 neurons” or “RS neurons/PCs” throughout. For instance, in the morphological analysis section, the term “RS PCs” appears, which may be confusing without consistent terminology.

R: We have revised the consistency of abbreviations. RS was defined at its first appearance (line 208 in revised manuscript), and RS neurons in the morphological analysis section were replaced by IT-2 neurons for consistency. Furthermore, abbreviations with low frequency, such as intrinsic burst (IB) and adaptive firing (AF), have been removed.

7) Discussion: the authors compare RMP values across studies and relate it to the age limitation (range 2–30 years). However, such comparisons are inherently limited: RMP measurements in whole-cell patch clamp are strongly influenced by recording conditions, including pipette and bath solutions, series resistance, dialysis effects, and the often-uncompensated liquid junction potential. Without standardized protocols, direct comparisons of absolute RMP values between studies are not reliable. Furthermore, the cited age range may not represent a major limitation. Barzó et al. (2025, eLife) analyzed a large dataset of human L2/3 pyramidal neurons (from cortical tissue close to pathological lesion – but in line also with non-human primate and rodent studies) across the lifespan and showed that the most pronounced changes in passive properties, including RMP, occur early in life (before age 1), with relatively stable values beyond that. This suggests that developmental shifts in RMP are unlikely to confound comparisons within the 2–30-year range, provided the data were recorded consistently.

R: We thank you for this observation. We acknowledge that RMP measurements are strongly influenced by the experimental recording conditions, as mentioned by the reviewer. We also appreciate that the reviewer brought up the work by Barzó et al. (2025), which allows us to better compare our data and refine the discussion narrative. However, we would like to highlight one observation from Barzó et al. While it is clear that the dramatic changes in passive membrane properties occur during the first year of life, subtle yet significant differences, including in RMP, can still be observed across life stages.

Although our study shows that, despite age differences (2–7 years, childhood, for the parietal cortex; 21–30 years, early adulthood, for the temporal and frontal cortices), RMP is relatively homogeneous across the regions examined, this remains an interesting observation. Barzó et al. (2025), using “non-pathological” or non-epileptic tissue, reported that RMP becomes significantly more hyperpolarized between early childhood and early adulthood (mean ± SD: –65.44 ± 6.82 mV vs. –68.69 ± 5.54 mV). While we recognize the difficulty of directly comparing RMP values across studies, the findings of Barzó et al. (2025), along with those cited in our discussion, support the notion that RMP values obtained from tissue derived from epileptic foci are more hyperpolarized than expected under non-epileptic conditions, possibly reflecting underlying pathological or compensatory mechanisms. In the revised manuscript, we have modified the paragraph in the discussion section to address this idea in the following paragraph (page 29):

Another important consideration is that electrophysiological measurements may vary across life stages (Barzó et al., 2025). For instance, the RMP in layer II/III PCs from non-epileptic tissue becomes more hyperpolarized from childhood to early adulthood (mean: –65 mV vs. –69 mV). In our study, despite age differences between the examined cortices (2–7 years, childhood, for the parietal cortex; 21–30 years, early adulthood, for the temporal and frontal cortices), the RMP was predominantly more hyperpolarized (–72 to –73 mV) than in cortical neurons from non-epileptic tissue (Moradi Chameh et al., 2021; Barzó et al., 2025). These observations support the idea that a more hyperpolarized RMP may be an underlying feature of epileptic tissue. However, it is worth noting that other factors, such as series resistance and liquid junction potential, can also influence RMP measurements.

8) Please, be consistent also with citation style as sometimes (Last name et al, year) is provided instead of numbers.

R: We have amended this mistake. Thanks.

Reviewer #3:

Review of “Layer V Neocortical Neurons From Individuals With Drug-Resistant Epilepsy Show Multiple Synaptic Alterations but Lack Somatic Hyperexcitability”

The authors performed whole-cell recordings from human layer-V pyramidal cells resected from the temporal, parietal, and frontal cortex in patients with drug-resistant epilepsy. Cells were classified as IT-2 (regular-spiking) and ET (intrinsic-bursting), with regional analyses primarily focused on IT-2 neurons. Using L I/II stimulation while recording in L Va, they measured presynaptic fiber volleys (FV) and field EPSPs. FV–fEPSP correlation was weak in human tissue but much stronger in a rodent control. A key finding is that 10-pulse, 30-Hz trains produced significant facilitation in the temporal cortex, whereas the frontal cortex showed only mild facilitation, and the parietal cortex showed none. These synaptic results were accompanied by more spontaneous postsynaptic events in the temporal cortex than in the other regions. In conclusion, the authors state that differences in synaptic properties, rather than cellular electrophysiology, underlie epilepsy.

The authors applied solid, classical methods and clearly state their aims. It is also important to enhance the human electrophysiological recording dataset, as done by the authors, rather than relying on a single dominant source of data. Nevertheless, several concerns arose during reading, some about the results themselves and others about the strength and framing of the conclusions. In some cases, the lack of proper control does not prevent publication but requires the authors to adjust their conclusion.

Major comments

• SSA and time from resection (Figure 4G).

SSA appears highly related to elapsed time after surgery/slicing (larger immediately post-resection). Did the authors record the time from surgery to experiment for each neuron and analyze SSA versus this interval?

R: We did not examine the relationship between the time from surgery and the timing of the analysis of spontaneous postsynaptic potentials shown in Figure 4. We would like to clarify that we measured SSA at the somatic level using whole-cell recordings (Figure 1) and spontaneous postsynaptic potentials elicited by gamma train stimulation (10 stimuli at 30 Hz) at the synaptic level using extracellular recordings (Figure 4). In both cases, the average time between brain tissue extraction during surgery and the beginning of electrophysiological recordings was 3 ± 0.5 h.

As we stated in our manuscript, the recorded neurons did not exhibit SSA at resting membrane potential in the whole-cell configuration. However, spontaneous postsynaptic potentials emerged during 30 Hz stimulation, which were highly correlated with the magnitude of facilitation in response to the 30 Hz train (Figure 4G), possibly enhancing excitatory drive during high-frequency periods of neuronal activity. Therefore, although it may seem logical that an earlier start of electrophysiological recordings would be associated with vigorous spontaneous postsynaptic activity, the consistent absence of SSA in whole-cell recordings and the controlled average delay between tissue extraction and the onset of experiments make such a relationship unlikely.

• Sag ratio (IT-2 vs ET).

Reported sag ratios (IT-2: 0.141 ± 0.072; ET: 0.164 ± 0.063) are smaller than expected from prior work and my own experience. Since rodents were also recorded, a rodent sag comparison could be informative. As it stands, the small difference raises concern that the ET vs IT-2 classification may be compromised.

R: Thanks for this observation. There are several factors that could explain this difference. First, methodological differences in measuring the sag ratio. For example, Kalmbach et al. (2021) reported sag values between 1.1 and 1.25, with ET neurons exhibiting a significantly higher sag ratio than IT-2 neurons. These values are considerably higher than our measurements. In our study, we calculated the sag ratio as the proportion between the voltage change from the maximal hyperpolarizing deflection to the steady-state voltage and the total voltage change from the resting membrane potential (RMP) to the maximal deflection in response to a −300 pA current injection, following a classical method for sag ratio estimation. Using the method described by Kalmbach et al., our measurements increased by approximately one unit, yielding values closer to those reported by Kalmbach et al.

Regarding the possible influence of sag ratio values on neuronal classification, we would like to clarify that the primary criterion for classifying neurons as ET or IT (type 1 and type 2 subtypes) was their firing pattern, specifically whether they exhibited intrinsic bursting or regular spiking. This parameter is considered the most reliable criterion for neuronal classification. That said, the absence of differences in sag ratio values observed between ET and IT-2 neurons are more likely explained by the tissue origin, which was derived from epileptic foci in patients with drug-resistant epilepsy, and by potential alterations in HCN channel expression, as briefly mentioned in the new section titled Electrophysiological Differences between IT-2 and ET neurons.

• Input resistance of ET neurons.

ET Rin of 122 ± 24.1 MΩ seems high relative to prior studies (around 50 MΩ). This calls into question the classification and could impact conclusions about IT-2 properties. Please reconcile with the literature and methods.

We understand this concern. The first criterion used to differentiate pyramidal neurons as IT-2 or ET neurons was their firing pattern—regular spiking or burst spiking, respectively. Once this classification was established, both groups displayed the classical differences in other electrophysiological parameters, such as spike adaptation, instantaneous firing frequency, and input resistance, which are among the most robust electrophysiological distinctions between these neuronal populations (Kalmbach et al., 2021). Regarding the higher input resistance values observed in our experiments compared with previous studies, several factors could contribute to these differences such as health status if the tissue (non-epileptic tissue or tissue distant from tumor zones vs tissue derived from epileptic foci), composition of cutting solution (NMDG-based vs. sucrose-based), the integrity of the dendritic tree after slicing process and levels of expression of HCN channels.

Given the reviewer’s timely concern about classifying pyramidal neurons as ET and IT-2, and the differences in the magnitude of classic electrophysiological hallmarks (input resistance and sag ratio), we added a section in discussion section addressing this point. In this section, we highlight the electrophysiological features that distinguish ET from IT-2 neurons and discuss possible explanations—such as characteristics related to tissue health or methodological differences—for the variations in input resistance and HCN-mediated sag. This section appears on pages 28-29.

• Rheobase.

Figures 2F and 2I appear inconsistent regarding rheobase. In Fig. 2I, the fitted curves suggest the smallest rheobase in parietal, not frontal, cortex. Fits in

---

## [Decision Letter · Decision Letter 2]

7 Jan 2026

Layer V Neocortical Neurons From Individuals With Drug-Resistant Epilepsy Show Multiple Synaptic Alterations but Lack Somatic Hyperexcitability

PONE-D-25-08682R2

Dear Dr. Galvan,

We’re pleased to inform you that your manuscript has been judged scientifically suitable for publication and will be formally accepted for publication once it meets all outstanding technical requirements.

Kind regards,

Michele Giugliano

Academic Editor

PLOS One

Additional Editor Comments (optional):

Reviewers' comments:

Reviewer's Responses to Questions

**Comments to the Author**

Reviewer #2: All comments have been addressed

2. Is the manuscript technically sound, and do the data support the conclusions?

Reviewer #2: Yes

3. Has the statistical analysis been performed appropriately and rigorously?

Reviewer #2: Yes

4. Have the authors made all data underlying the findings in their manuscript fully available?

Reviewer #2: Yes

5. Is the manuscript presented in an intelligible fashion and written in standard English?

Reviewer #2: Yes

Reviewer #2: (No Response)

**Do you want your identity to be public for this peer review?** For information about this choice, including consent withdrawal, please see our Privacy Policy

Reviewer #2: **Yes:** Laura Monni

---

## [Editor Report · Acceptance letter]

PONE-D-25-08682R2

PLOS One

Dear Dr. Galvan,

I'm pleased to inform you that your manuscript has been deemed suitable for publication in PLOS One. Congratulations! Your manuscript is now being handed over to our production team.

Kind regards,

on behalf of

Dr. Michele Giugliano

Academic Editor

PLOS One